# GraphMP: Graph Neural Network-based Motion Planning with Efficient Graph Search

**Xiao Zang**[1]    **Miao Yin**[2*]    **Jinqi Xiao**[1]    **Saman Zonouz**[3]    **Bo Yuan**[1]

[1]Department of Electrical and Computer Engineering, Rutgers University
[2]Department of Computer Science and Engineering, The University of Texas at Arlington
[3]School of Cybersecurity and Privacy, Georgia Institute of Technology
[1]{xz514, jx257, bo.yuan.ece}@rutgers.edu
[2]miao.yin@uta.edu    [3]szonouz6@gatech.edu

## Abstract

Motion planning, which aims to find a high-quality collision-free path in the configuration space, is a fundamental task in robotic systems. Recently, learning-based motion planners, especially the graph neural network-powered, have shown promising planning performance. However, though the state-of-the-art GNN planner can efficiently extract and learn graph information, its inherent mechanism is not well suited for graph search process, hindering its further performance improvement. To address this challenge and fully unleash the potential of GNN in motion planning, this paper proposes GraphMP, a neural motion planner for both low and high-dimensional planning tasks. With the customized model architecture and training mechanism design, GraphMP can simultaneously perform efficient graph pattern extraction and graph search processing, leading to strong planning performance. Experiments on a variety of environments, ranging from 2D Maze to 14D dual KUKA robotic arm, show that our proposed GraphMP achieves significant improvement on path quality and planning speed over state-of-the-art learning-based and classical planners; while preserving competitive success rate.

## 1    Introduction

Motion planning aims to find a high-quality collision-free path connecting the start and goal states in the configuration space of a robot. As a fundamental cognitive task in robotic systems, motion planning plays a critical role in many practical applications, such as autonomous driving, in-warehouse package handling and assisted surgery, etc. A motion planning problem can be solved from different perspectives. Sampling-based solutions, e.g., RRT and its variants [1, 2] randomly sample the configuration space to build a space-filling tree, which grows towards connecting the start and goal configurations. Search-based approaches, such as A* and Dijkstra [3, 4], interpret the planning as a graph search problem, and they then find the feasible path via traversing the graph.

Recently, learning-based planners have obtained substantial attention because of their comparable or superior performance to classical planners. In general, such data-driven strategy can efficiently learn the patterns in the configuration space and/or the behaviors of the oracle planners, optimizing the important operation (e.g., sampling mechanism) in the planning process, and thus reducing the unnecessary collision check with the improved path quality. To date, various neural network-powered motion planners, including multilayer perceptron (MLP)-based [5], convolutional neural networks (CNN)-based [6], recurrent neural network (RNN)-based [7] and graph neural network (GNN)-based [8, 9], have been proposed in the literature. Among these different choices, the GNN-based neural planner is the most promising approach because of its unique advantages in processing

---

[*]This work was done when the author was with Rutgers University.

37th Conference on Neural Information Processing Systems (NeurIPS 2023).

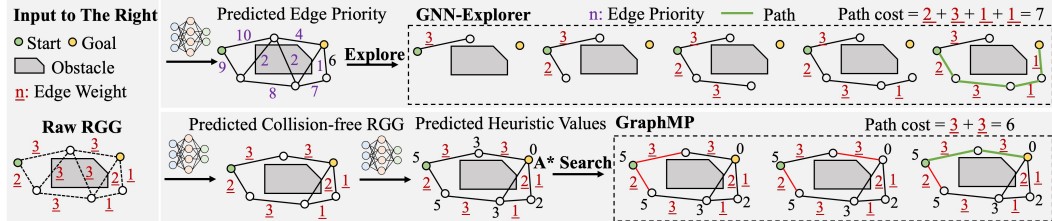

Figure 1: The comparison between the key ideas of GNN-Explorer and our GraphMP. The finally identified paths are marked as green lines. The GNN-Explorer first predicts the exploration priority of all edges and then incrementally expands the exploration tree with collision-free edges in sorted order. On the other hand, our GraphMP predicts both the collision status of edges and the heuristic values of all nodes using two neural networks, and utilizes the edge weights to perform the A* search. For each step of the A* search, we mark the connection to the candidate nodes with red lines. It is seen that our method fully leverages the information of real accumulated costs during the search procedure, leading to a higher quality of the searched path.

challenging high-dimensional planning tasks. By sampling the continuous configuration space and generating the random geometric graph (RGG), the equipped GNN, which is naturally powerful for learning graph patterns, can be applied to predict the priority of nodes (configurations) [9] or edges (robot movement) [8], improving the efficiency and quality of path exploration. In particular, GNN- Explorer [8], as the most recent and representative GNN planner, has demonstrated superior performance in a series of low and high-dimensional planning tasks than various learning-based and classical planners.

However, we believe the potential of GNN-based motion planners is not fully unleashed and the state-of-the-art solution can be further improved. Our key observation is that the edge priority-based path exploration, by its nature, is not the optimal choice for solving graph search problems. For instance, as illustrated in Figure 1, neither the edge priority generation nor the path construction process fully considers the impact on the total path cost when selecting a new edge to the current path. In other words, the critical information of the accumulated cost is not properly represented, leveraged or maintained in the existing GNN-based planners, thereby limiting the quality of the finally searched path. On the other hand, the classical graph search approaches, such as A*, can provide the near-optimal solution for finding the shortest path over the weighted graph, if the suitable heuristic function can be identified. Essentially, the key factors for the success of A*-like approaches in the graph search task are 1) they fully explore the graph structure via routinely visiting the neighboring nodes; and 2) they formally consider the impact of the accumulated cost during path exploration; and these two features are exactly what the existing GNN and most of other learning-based planners lack.

Motivated by these insights, in this paper we propose to simultaneously leverage the advantages of GNN and graph search algorithm, leading to an efficient neural motion planner, namely GraphMP. GraphMP first effectively extracts and learns the important patterns in the configuration space via its GNN modules, and then identifies the near-optimal path over the processed RGG using its low-cost graph search component. More specifically, to enable this processing pipeline, a GNN-based neural collision checker and a neural heuristic estimator are proposed to extract key graph information from input RGG and provide it to A* module, which is also reformulated in the differentiable way to facilitate end-to-end training. More details of GraphMP is shown in Figure 2 and described in Section 3. We evaluate GraphMP and other learning-based and classical planners in a variety of environments, ranging from 2D Maze to 14D dual KUKA robotic arm. Experimental results show that our proposed GraphMP achieves significant improvement on various planning performance metrics (i.e., path quality and planning speed) over the state-of-the-art planning methods; while preserving the competitive success rate.

## 2 Related Work

**Classical Planners.** Today's most popular motion planners are either search-based or sampling-based. *Search-based planners*, including BFS [10], Dijkstra [4], and A* [3], are typically used for finding optimal paths in low-dimensional environments. Different from BFS and Dijkstra that are significantly slow in large maps, A* is an informed search algorithm that uses a heuristic function to guide the search towards the goal, significantly reducing the search space and improving the search efficiency.

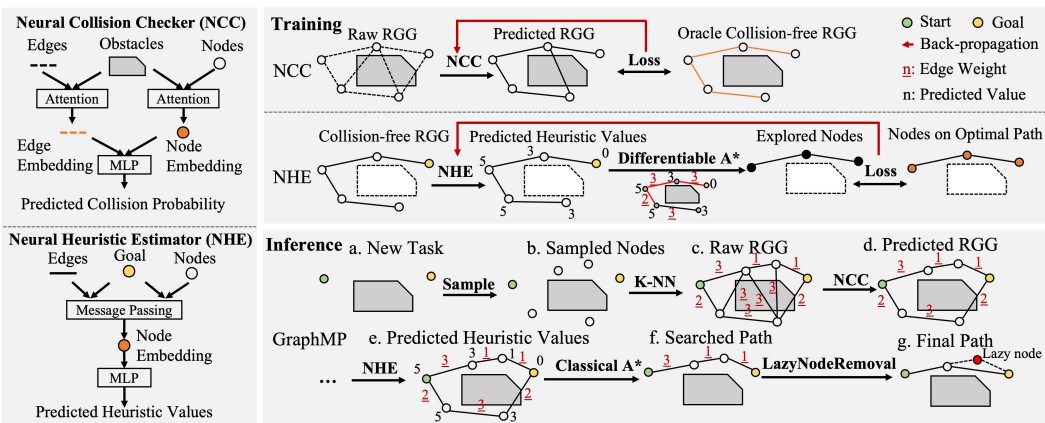

Figure 2: **(Left)**: The architectures of neural collision checker (**NCC**) and neural heuristic estimator (**NHE**). The neural collision checker takes a raw RGG and obstacles as input and predicts the collision status of all edges. The neural heuristic estimator takes the collision-free RGG and the goal as input and predicts the heuristic values of all nodes. **(Top right)**: The independent training phases of NCC and NHE. Notice that here we design and utilize a differentiable graph-based A* concatenated to NHE, enabling their joint training in an end-to-end manner. **(Bottom right)**: The main steps of the inference phase to solve the planning task. Once a valid solution is found, we perform the lazy node removal to further reduce the path cost.

Motivated by this benefit, many variants of A* have been developed, such as D* [11], WA* [12] and Hybrid-A* [13]. The heuristic functions adopted in these methods are typically admissible hand-craft versions, such as Euclidean distance and Manhattan distance.

Unlike search-based planners, *sampling-based planners* probe the high-dimensional configuration space with random sampling, alleviating the exponential complexity growth incurred by the increasing number of dimensions. In general, this type of planner first constructs a randomly sampled graph, and then explores the feasible path over the sampled graph. One important sub-category of sampling-based planner is tree-based solution, including RRT [1], RRT* [14] and Informed-RRT* [15], etc. Some other popularly used sampling-based planners include PRM [16], BIT* [2] and LazySP [17].

**Learning-based Planners.** Recently, a series of learning-based methods further extend the search-based planners by producing better heuristics to guide the search. For instance, SAIL [18] trains heuristic policies by imitating clairvoyant oracles and demonstrates the capability of reducing search effort. [19] and [20] learn the heuristic functions using the U-Net [21] and transformer [22] architecture, respectively. On the other hand, VIN [23] and SPT [24] learn the planning policy directly, and [25] develops the differentiable solvers for integer linear optimization problems by treating them as black boxes. Furthermore, Neural Weighted A* [26] learns the image-format graph costs and heuristics simultaneously via being supervised under the planning examples. Neural A* [27] proposes to reformulate the entire procedure of the canonical A* search algorithm to be differentiable, enabling the end-to-ending training of a guidance map for 2-D path planning tasks. However, this method still relies on a hand-crafted heuristic function and can only be applied to 2D images, hindering its applications.

In addition, a variety of deep learning-powered sampling-based planners have also been proposed in the literature. Fastron [28] and ClearanceNet [29] learn the function approximators to facilitate collision detection. On the other hand, LEGO [30] and [31] predict the high-quality sampling distribution using conditional variational auto-encoder (CVAE), reducing the exploration effort. MPNet [5] and STP-Net [7] predict the next sample directly by encoding the environments together with the configurations via the contractive auto-encoder (CAE) and convolutional layers, respectively. In particular, motivated by their powerful capability of learning the topology of environments of any dimension, the application of GNNs in motion planning has gained popularity in recent years. [9] introduces a GNN-based sampler to predict the critical node in a randomly generated graph, showing promising performance for identifying the optimal samples in high-dimensional tasks. [8] proposes GNN-Explorer, which prioritizes the exploration of graph edges in the planning process. Since the planner can now focus on the most promising paths first, the planning efficiency of GNN-Explorer is significantly improved, outperforming various classical and learning-based planners.

## 3 Our Approach: GraphMP

### 3.1 Preliminaries

Consider a motion planning problem defined in the $d$-dimensional continuous configuration space $\mathcal{X}$, where $\mathcal{X}_{obs}$ is the obstacle space and $\mathcal{X}_{free} = \mathcal{X} \setminus \mathcal{X}_{obs}$ is the free space. Given an RGG $\mathcal{G} = (\mathcal{V}, \mathcal{E})$ $\mathcal{V}$, where $\mathcal{V}$ is the set of nodes sampled from $\mathcal{X}_{free}$ and $\mathcal{E}$ is the set of weighted edges constructed by K-nearest neighbors (KNN). A motion planner aims to find a collision-free path that connects the source node $v_s$ and goal node $v_g$ by a collision-free path. The prior work GNN-Explorer [8] initializes an exploration tree $\mathcal{T}$ rooted from $v_s$, and visits edges $e \in \mathcal{E}$ sorted by their predicted priorities. Each visit performs the accurate collision check on the edge and tries to append it into $\mathcal{T}$, until $v_g$ is reached. Here the search of GNN-Explorer relies on the predicted edge priorities without fully exploring graph structure and formally consider the impact of accumulated path cost during the path exploration, thereby hampering the path quality. On the other hand, graph-based A* computes the path from $v_s$ by iteratively selecting the best candidate $v_{sel} = \arg\min_{v \in \mathcal{O}}(g(v) + f(v))$ from the open list $\mathcal{O}$. Here, $g(v)$ accumulates the actual cost from $v_s$ to $v$ and $f(v)$ estimates the heuristic value of the cost from $v$ to $v_g$. Once $v_{sel}$ is selected, each of ites reachable neighboring nodes $v_{nbr} \in \mathcal{V}_{sel}$ is checked and updated. By using this way, A* can better explore graph stcuture and perform cost-aware search, but it requires manual design of high-quality heuristic function $h(v)$, which is a challenging task for high-dimensional tasks. Also, visiting the neighboring nodes can be time-consuming because the accurate collision check must be performed on each edge $e_{v_{sel}, v_{nbr}}$. To overcome these limitations of prior works, GraphMP proposes to use GNN to extract and learn the important patterns of RGG, and then identifies the near-optimal path using learnable graph search component. More specifically, a neural collision checker and a neural heuristic estimator are proposed to extract key graph information from input RGG and provide it to the proposed reformulated differentiable A* module for end-to-end training. Therefore, the path planning is now a graph structure-aware and cost-aware process with powerful graph pattern extraction capability and learnable heuristics function, making it can achieve high planning performance and suitable for both low and high-dimensional tasks.

### 3.2 Overall Framework

Fig. 2 shows the overall architecture of the proposed GraphMP. In the **training phase**, a *neural collision checker* takes the raw RGG and obstacle information as input and predicts the collision status of all edges. The difference between the estimated and ground-truth collision-free RGGs serves as the training loss to improve the prediction quality of the neural collision checker. Meanwhile, a *neural heuristic estimator* is also individually trained to assign the proper heuristic value to each node in the ground-truth collision-free RGG, making the graph search-based planning become possible. Here in order to make the neural heuristic estimator learnable, its predicted heuristic values are sent to A* module to generate a list of nodes that should be visited. The difference between this predicted node list and the optimal path is then calculated as the training loss, guiding the update of neural heuristic estimator. Notice that the A* module in the training phase is designed in a *differentiable* way to enable the backward propagation of gradients. More details will be described in Section 3.5.

After individually training neural collision checker and neural heuristic estimator, these two modules are then concatenated in the **inference phase**. More specifically, neural collision checker first predicts the potential collisions existed in the RGG and generates a predicted collision-free RGG. The neural heuristic estimator then assigns the estimated heuristic value to each node of this roughly collision-free RGG, enabling the graph search-based planning via a A* module. Notice that because in the inference phase the input RRG of A* is only approximately collision-free without guarantee, a non-differentiable classical A* is adopted here to incorporate the potentially additional collision check during the search procedure. After the nodes of a collision-free path are finally identified, a lazy node removal (details in Section 3.5) is performed to further improve path quality.

### 3.3 Neural Collision Checker

Because detecting the potential collision of an edge, by it nature, can be interpreted as a binary classification problem, we build a neural network-based collision checker to roughly predict the collision status of the edge $e_{ij}$ in the input RGG $G = (\mathcal{V}, \mathcal{E})$ with obstacle information $\mathcal{X}_{obs}$:

$$e_{ij} \text{ is collision-free} = \begin{cases} \textbf{True} & \text{if } p_{\mathcal{I}(e_{ij})} > 0.5 \\ \textbf{False} & \text{otherwise,} \end{cases} \tag{1}$$

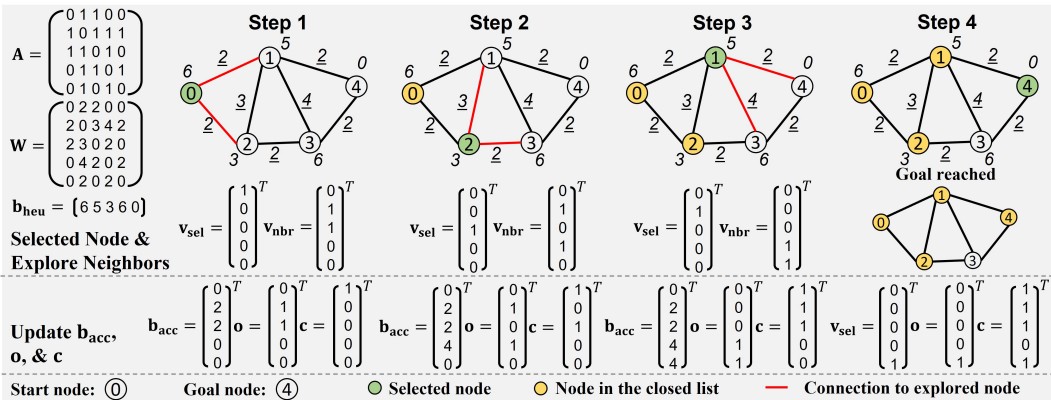

Figure 3: An example of performing the differentiable graph-based A* to find a path from node 0 to node 4. Given a weighted graph, $\mathbf{A}$, $\mathbf{W}$ and $\mathbf{b_{heu}}$ are the corresponding binary adjacency matrix, weighted adjacency matrix and the heuristic values, respectively. At each step, we first compute the lowest-cost candidate $\mathbf{v_{sel}}$ according to Eq. 3 and explore its neighboring nodes $\mathbf{v_{nbr}}$ via Eq. 5. The accumulated cost $\mathbf{b_{acc}}$ is updated via Eq. 7. Besides, we keep track of the nodes to be visited and have been visited by updating $\mathbf{o}$ and $\mathbf{c}$ via Eq. 4 and Eq. 8. Once the goal is reached, all the visited nodes are recorded in the closed list vector $\mathbf{c}$.

where $\mathcal{I}(e_{ij})$ is the function that gives the index of $e_{ij}$ in the edge set $\mathcal{E}$, and $\mathbf{p} = (p_0, p_1, ...p_{|\mathcal{E}|-1})$ is the *collision-free probability* vector that can be calculated as: $\mathbf{p} = f_{prob}(f_{oe}(\mathcal{V}, \mathcal{E}, \mathcal{X}_{obs}))$. Here $f_{oe}$ is the attention-based iterative obstacle encoder [8] that encodes the information of input RGG and obstacles to a concatenated embedding vector $\mathbf{u} = (x_i^{(l)}, x_j^{(l)}, x_i^{(l)} - x_j^{(l)}, y_{ij}^{(l)})$ for each edge $e_{ij}$, where $x_i^{(l)}$ and $y_{ij}^{(l)}$ denote the node embedding of $v_i$ and edge embedding of $e_{ij}$ at the $l$-th iteration of obstacle encoding, respectively. $\mathbf{u}$ is then processed by a three-layer MLP $f_{prob}$ to calculate the collision-free probability for each edge $e_{ij}$. More details of $f_{oe}$ and $f_{prob}$ are in the Appendix.

**Training Procedure.** In the training phase the neural collision checker is individually trained on batches of collision check problem instances $\{(\mathcal{V}^{(i)}, \mathcal{E}^{(i)}), \mathcal{X}_{obs}^{(i)}\}$, where the collision-free probabilities of all the collision-free edges in the $i$-th problem instance is set as 1, otherwise 0. The accurate labeling on the training data is provided by the oracle collision checker. The difference between the estimated collision-free probability vector $\mathbf{p}$ and the binary representation of the ground-truth label, is measured via calculating binary cross-entropy (BCE) loss and minimized during the training procedure to improve the prediction quality of neural collision checker.

## 3.4 Neural Heuristic Estimator

Given the input collision-free RGG $\mathcal{G}_{free} = (\mathcal{V}, \mathcal{E}_{free})$ and goal node $v_g$, the neural heuristic estimator is constructed as a graph neural network (GNN)-based module to predict the heuristic value $b_{heu,i}$ for each $v_i \in \mathcal{V}$. To that end, two MLPs (detailed in the Appendix) are first used to embed the information of $v_i$ and $e_{ij}$ to latent space as $q_i^{(0)} \in \mathbb{R}^{d_h}$ and $r_{ij}^{(0)} \in \mathbb{R}^{d_h}$, respectively, where $d_h$ is the embedding size, and then the GNN iteratively updates the node and edge embeddings via aggregating the local information of each node from its neighbors $\mathcal{N}(v_i) = \{v_j | e_{ij} \in \mathcal{E}_{free}\}$ as follows:

$$m_i^{(l)} = \max(\{f_q(q_i^{(l)}, q_j^{(l)}, q_j^{(l)} - q_i^{(l)}, r_{ij}^{(l)}) | v_j \in \mathcal{N}(v_i)\}), \quad q_i^{(l+1)} = g(q_i^{(l)}, m_i^{(l)}), \forall v_i \in \mathcal{V},$$
$$r_{ij}^{(l+1)} = \max(r_{ij}^{(l)}, f_r(q_i^{(l)}, q_j^{(l)}, q_j^{(l)} - q_i^{(l)})), \forall e_{ij} \in \mathcal{E}_{free}. \tag{2}$$

Here, $g$, $f_q$ and $f_r$ are two-layer MLPs with output dimension as $d_h$. After $L$ iterations, the heuristic value of node $i$ is calculated via encoding the node embedding $q_i^{(L)}$ as $b_{heu,i} = f_{val}(q_i^{(L)})$, where $f_{val}$ is a three-layer MLP.

## 3.5 Differentiable Graph-based A*

In order to ensure that the neural heuristic estimator can provide good estimation of heuristic value for graph search-based planning, in the training phase it is concatenated to the A* module, allowing the final planning result to directly guide the training procedure. However, a challenging issue for enabling such end-to-end learning is the non-differentiability of A*, incurred by the discrete nature of its incremental search procedure. Notice that though [27] proposes a differentiable grid-based A*, this existing solution can only work for 2-D grid search and planning; while addressing the non-differentiability of the more general graph-based A*, which is used for our proposed GraphMP towards high-dimensional planning, is non-trivial and not explored yet.

Next we describe the key reformulation for differentiable graph-based A*, which is illustrated in Fig. 3. Recall that the essence of A* on $\mathcal{G}_{free} = (\mathcal{V}, \mathcal{E}_{free})$ is to iteratively select and move the node associated with the lowest *path cost* from open list $\mathcal{O}$, which contains all the currently candidate nodes, to the closed list $\mathcal{C}$ that stores the visited nodes, and explore the neighbors of the selected nodes to expand $\mathcal{O}$ (More details of A* are described in the Appendix). To make this procedure differentiable, we first use two binary vectors $\mathbf{o} \in [0,1]^{|\mathcal{V}|}$ and $\mathbf{c} \in [0,1]^{|\mathcal{V}|}$ to represent the nodes information in $\mathcal{O}$ and $\mathcal{C}$, respectively, where the 1 entries indicate the contained nodes. Also, the *accumulated costs* and the estimated heuristic values of all the nodes in $\mathcal{V}$ are denoted as real-valued $\mathbf{b_{acc}} = (b_{acc,0}, b_{acc,1}, ..., b_{acc,|\mathcal{V}|-1})$ and $\mathbf{b_{heu}} = (b_{heu,0}, b_{heu,1}, ..., b_{heu,|\mathcal{V}|-1})$, respectively, where $b_{heu,i}$ is predicted by the neural heuristic estimator. Then, the operations of selecting the lowest-cost node in $\mathcal{O}$ and moving it from $\mathcal{O}$ to $\mathcal{C}$ can be vectorized via using a one-hot vector $\mathbf{v_{sel}} \in [0,1]^{|\mathcal{V}|}$ as follows:

$$\mathbf{v_{sel}} = \mathcal{I}_{\mathbf{max}}\left(\frac{\exp(-(\mathbf{b_{acc}} + \mathbf{b_{heu}})/\lambda) \odot \mathbf{o}}{\exp(-(\mathbf{b_{acc}} + \mathbf{b_{heu}})/\lambda)\mathbf{o}}\right), \tag{3}$$

$$\mathbf{o} = \mathbf{o} - \mathbf{v_{sel}}, \mathbf{c} = \mathbf{c} + \mathbf{v_{sel}}. \tag{4}$$

Here similar to [27], element-wise product $\odot$ is used to mask the nodes in $\mathcal{C}$, and $\lambda$ and $\mathcal{I}_{\mathbf{max}}(.)$ are the pre-set parameter and the function returns a one-hot vector with the "1" entry associated with the lowest path cost $(b_{acc,i} + b_{heu,i})$. $\mathbf{v_{sel}}$ is then further used to vectorize the neighborhood exploration process. To be specific, let $\mathbf{A} \in \{0,1\}^{|\mathcal{V}| \times |\mathcal{V}|}$ and $\mathbf{W} \in \mathbb{R}^{|\mathcal{V}| \times |\mathcal{V}|}$ denote the unweighted and the weighted adjacency matrix of $\mathcal{E}_{free}$, respectively. Accordingly, the neighboring nodes that should be explored can be represented as

$$\mathbf{v_{nbr}} = \mathbf{A}\mathbf{v_{sel}} \odot (\mathbb{1} - \mathbf{c}), \tag{5}$$

where $\mathbf{v_{nbr}} \in [0,1]^{|\mathcal{V}|}$ is the binary vector marking the entries corresponding to the neighboring nodes as ones, and $\mathbb{1}$ is the all-one vector. Notice that here the indices of the neighboring nodes that are already in the closed list $\mathcal{C}$ are masked out via using $(\mathbb{1} - \mathbf{c})$. Upon the identification of these newly explored neighboring nodes, the accumulated costs $\mathbf{g}$ and the vectorized open list $\mathbf{o}$ are further updated as:

$$\mathbf{b'_{acc}} = \mathbf{b_{acc}} \odot \mathbf{v_{sel}} + \mathbf{W}\mathbf{v_{sel}}, \tag{6}$$

$$\mathbf{\Phi} = ((\mathbb{1} - \mathbf{o}) + \mathbf{o} \odot (\mathbf{b_{acc}} > \mathbf{b'_{acc}})) \odot \mathbf{v_{nbr}}, \ \mathbf{b_{acc}} = \mathbf{b_{acc}} \odot (\mathbb{1} - \mathbf{\Phi}) + \mathbf{b'_{acc}} \odot \mathbf{\Phi}, \tag{7}$$

$$\mathbf{o} = \mathbf{o} + (\mathbb{1} - \mathbf{o})\mathbf{v_{nbr}}, \tag{8}$$

where $\mathbf{b'_{acc}}$ denotes the accumulated costs of the nodes that are associated the path containing the selected node. The operation $\mathbf{b_{acc}} > \mathbf{b'_{acc}}$ compares the node-wise accumulated costs and yields a binary vector of length $|\mathcal{V}|$. More specifically, $\mathbf{b_{acc}} \odot \mathbf{v_{sel}}$ extracts the accumulated cost of the currently selected node, and $\mathbf{W}\mathbf{v_{sel}}$ represents the distance from the selected node to each of its neighbors. Notice that as shown in Eq. 7, the accumulated cost of neighboring nodes are updated in two cases: (1) the nodes do not exist in $\mathcal{O}$; and (2) the nodes are in $\mathcal{O}$ but its current accumulated cost is already smaller than the updated one. The identification of such two types of nodes are realized by a binary vector $\mathbf{\Phi}$. After updating the accumulated costs, the open list $\mathcal{O}$ is expanded by adding newly explored neighboring nodes (Eq. 8).

**End-to-End Training.** With the differentiability of graph-based A*, neural heuristic estimator and graph search planning can be now jointly trained in an end-to-end way, providing better guidance of learning heuristic values. Algorithm 1 describes the overall training procedure.

**Algorithm 1** End-to-End Training Framework

**Input:** Full training set $\mathcal{D}_{heu}$, neural heuristic estimator $f_{heu}(\mathcal{V}, \mathcal{E}, v_g, \mathbf{\Theta_{heu}})$ with weights $\mathbf{\Theta_{heu}}$, the max iteration $T_{max}$, learning rate $\gamma$

1: **for** $(\mathcal{V}^{(i)}, \mathcal{E}_{free}^{(i)}, v_s^{(i)}, v_g^{(i)}, \hat{\mathbf{c}}^{(i)})$ in $\mathcal{D}_{heu}$ **do**
2: $\quad$ Predict $\mathbf{b_{heu}} \leftarrow f_{heu}(\mathcal{V}^{(i)}, \mathcal{E}_{free}^{(i)}, v_g^{(i)}, \mathbf{\Theta_{heu}})$. # Predict the heuristic values.
3: $\quad$ $\mathbf{b_{acc}} \leftarrow \mathbf{0}, \mathbf{o} \leftarrow \mathbf{v}_s, \mathbf{c} \leftarrow \mathbf{0}$.
4: $\quad$ **for** $t = 1, 2, ..., T_{max}$ **do**
5: $\quad\quad$ Select $\mathbf{v_{sel}}$ via Eq. 3.
6: $\quad\quad$ **if** $\mathbf{v_{sel}} == \mathbf{v_g}$ **then Break** # $\mathbf{v_g}$ is the one-hot vector representing the goal $v_g^{(i)}$.
7: $\quad\quad$ **if** $sum(\mathbf{o}) == 0$ **then Break** # Early stopping when the open list is empty.
8: $\quad\quad$ $\mathbf{o} \leftarrow \mathbf{o} - \mathbf{v_{sel}}, \mathbf{c} \leftarrow \mathbf{c} + \mathbf{v_{sel}}$.
9: $\quad\quad$ $\mathbf{v_{nbr}} \leftarrow \mathbf{A}\mathbf{v_{sel}} \odot (\mathbb{1} - \mathbf{c})$.
10: $\quad\quad$ $\mathbf{b_{acc}} \leftarrow \mathbf{b_{acc}} \odot (\mathbb{1} - \mathbf{\Phi}) + \mathbf{b_{acc}'} \odot \mathbf{\Phi}$ (Eq. 6-7).
11: $\quad\quad$ $\mathbf{o} \leftarrow \mathbf{o} + (\mathbb{1} - \mathbf{o})\mathbf{v_{nbr}}$.
12: $\quad$ **end for**
13: $\quad$ # $\hat{\mathbf{c}}^{(i)}$ is a length-$|\mathcal{V}|$ binary vector that marks all nodes along the optimal path as ones.
14: $\quad$ Compute the loss $\mathcal{L}_{heu} \leftarrow ||\hat{\mathbf{c}}^{(i)}, \mathbf{c}||_1$.
15: $\quad$ Update weights $\mathbf{\Theta_{heu}} \leftarrow \mathbf{\Theta_{heu}} - \gamma\nabla_{\mathbf{\Theta_{heu}}}\mathcal{L}_{heu}$.
16: **end for**

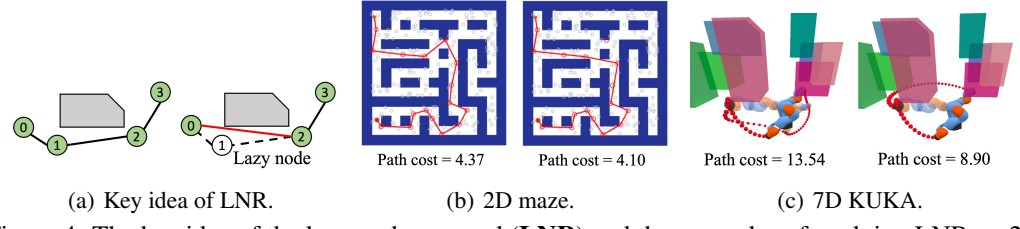

| (a) Key idea of LNR. | (b) 2D maze. | (c) 7D KUKA. |

Figure 4: The key idea of the lazy node removal (**LNR**) and the examples of applying LNR on 2D maze and 7D KUKA environments. We compare the paths before LNR (left) and after LNR (right).

## 3.6 Inference Procedure for Online Planning

As illustrated in Fig. 2, when GraphMP performs online planning (detailed in the Appendix) for the new task, a non-differentiable classical A* module is adopted to perform graph search on the approximated collision-free RGG prepared by the neural collision checker. Here the heuristic values desired in the search procedure is predicted by the neural heuristic estimator. Notice that in order to reduce the computational cost and path cost, the following two optimization operations are adopted in the inference procedure.

**In-Search Collision Check.** Due to the approximation property of neural networks, some edges in the approximated collision-free RGG may collide with the obstacles. We integrate the accurate collision check into A* search that only occurs on necessary edges with low prediction confidence. specifically, we perform the time-consuming accurate collision check only when (1) the neighbor of one selected node is explored, and (2) the collision-free probability of the corresponding edge is lower than a threshold $\theta$. Such operation balances the trade-off between the planning efficiency and the path safety.

**Lazy Node Removal.** After A* finishes the path search, lazy node removal is performed to further reduce path cost. As illustrated in Fig. 4, *within the planned path*, the pairs of the nodes that are not directly connected are iteratively checked to see whether if the potential direct connection (as new edge) is collision-free and brings shorter path. If so, the two nodes can be connected and the nodes between them are removed as "lazy nodes". In general, the worst time complexity of lazy node removal is $O(T^2)$, where $T$ is the number of nodes in the path found by A*. Since the searched path typically contains only few nodes ($T$ is small), this technique effectively reduces the path cost with minor computing overhead.

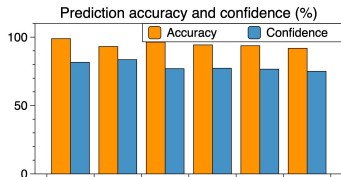
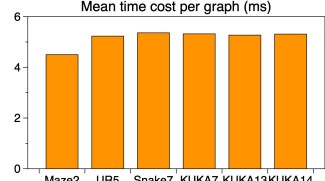
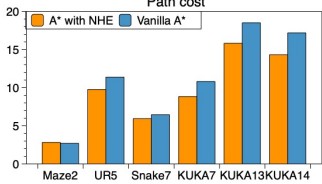

| (a) Performance of NCC. | (b) Performance of NHE. |

Figure 5: **(Left)**: The performance of neural collision checker (NCC) on 1000 raw RGGs with 300 nodes and $K$-value of 10 (K-NN). **(Right)**: The comparison between A* with the heuristic function of neural heuristic estimator (NHE) and vanilla A*, with respect to the mean path cost. More numerical results are in the Appendix.

## 4 Results

### 4.1 Dataset and Experimental Setup

We demonstrate the effectiveness of our algorithm in six types of planning tasks as the same in [8]: (1) Maze2: a 2 DoF point-robot in 2D maze workspace of dense obstacles, (2) Ur5: a 6 DoF UR5 robot in 3D workspace, (3) Snake7: a 7 DoF snake robot in 2D workspace, (4) Kuka7: a 7DoF KUKA arm in 3D workspace, (5) Kuka13: a 13 DoF Kuka arm in 3D workspace, and (6) Kuka14: a pair of 7DoF KUKA arms in 3D workspace.

For each environment, we prepare two different training datasets each of which consists of 2000 different workspaces, to train the neural collision checker and heuristic estimator separately. For each workspace, we randomly construct 20 RGGs by sampling a random number of nodes ([100, 200, 300, 400]) and a random value of KNN ([5, 10, 15, 20]). Specifically, for the neural collision checker, each problem contains the raw RGG built on a set of randomly sampled free states and the different set of obstacles, while the ground-truth is the collision-free RGG obtained by the oracle collision checker. For the neural heuristic estimator, each problem contains a feasible pair of $v_s$ and $v_g$, the exactly collision-free graph, and the ground-truth is the optimal path computed by Dijkstra. We also use 500 problem instances as the validation set. We keep the weights of heurisitic estimator which yields the smallest average path cost on the validation set. After the training, GraphMP is evaluated in an end-to-end manner on 1000 problem instances with unseen workspaces. Each input of the testing problems contains a pair of $v_s$ and $v_g$, and a set of obstacles.

Our neural collision checker adopts 3 iterations of obstacle encoding with an output dimension of 64, and the neural heuristic estimator has 5 loops of message passing in Eq. 2 with the output dimension of 32. For both training of these two models, we select ADAM [32] as the optimizer and set the learning rate as $1e^{-3}$. The training epoch is 400 and the batch size is set as 8. We set the threshold $\theta$ in the in-search collision check as 80%. The number of graph nodes per sampling is 100 and the $K$ value of K-NN is 10. The experiments are conducted on a computere quipped with an AMDEPYC 74202P24-Core Processor and an NVIDIA RTXA6000 GPU.

**Baselines.** We evaluate the performance of GraphMP by comparing it with three classical planners (BIT*, RRT* and LazySP) and one state-of-the-art learning-based planner (GNN-Explorer). In addition, the Smoother proposed in [8] serves as an additional step to reduce the path cost of the

Table 1: The overall success rate on 1000 testing problems for each environment. GraphMP archives nearly 100% success rate across all environments, being competitive with other baseline planners.

|  | Maze2 | UR5 | Snake7 | KUKA7 | KUKA13 | KUKA14 |
|---|---|---|---|---|---|---|
| BIT* | 1.00 | 1.00 | 1.00 | 1.00 | 1.00 | 1.00 |
| RRT* | 0.54 | 0.39 | 0.69 | 0.83 | 0.67 | 0.70 |
| LazySP | 1.00 | 0.99 | 1.00 | 1.00 | 0.99 | 0.99 |
| **GraphMP** | 1.00 | 0.99 | 1.00 | 0.99 | 1.00 | 0.99 |
| GNN-Explorer | 1.00 | 0.98 | 1.00 | 0.99 | 1.00 | 0.99 |
| **GraphMP with Smoother** | 1.00 | 0.99 | 1.00 | 0.99 | 1.00 | 0.99 |
| GNN-Explorer with Smoother | 1.00 | 0.98 | 1.00 | 0.99 | 1.00 | 0.99 |

Table 2: Mean path cost on 1000 testing problems for each environment. GraphMP with Smoother achieves the lowest path cost across all environments, compared to the baselines with a similar success rate. (The success rate of RRT* on Maze2 is 54%.)

|  | Maze2 | UR5 | Snake7 | KUKA7 | KUKA13 | KUKA14 |
|---|---|---|---|---|---|---|
| BIT* | 2.52 | 11.20 | 5.96 | 7.59 | 12.07 | 12.05 |
| RRT* | **1.82** | 10.35 | 5.06 | 7.07 | 10.03 | 10.65 |
| LazySP | 2.63 | 11.81 | 6.52 | 9.49 | 16.72 | 16.82 |
| **GraphMP** | 2.33 | 8.09 | 5.68 | 7.01 | 13.57 | 12.22 |
| GNN-Explorer | 2.80 | 12.61 | 6.50 | 9.15 | 16.75 | 16.50 |
| **GraphMP with Smoother** | 1.96 | **7.73** | **5.02** | **6.27** | **9.26** | **9.89** |
| GNN-Explorer with Smoother | 2.36 | 8.87 | 5.31 | 6.55 | 9.92 | 10.01 |

Table 3: The mean time cost (ms). GraphMP demonstrates the fastest planning speed, among 4 of 6 environments.

|  | Maze2 | UR5 | Snake7 | KUKA7 | KUKA13 | KUKA14 |
|---|---|---|---|---|---|---|
| BIT* | 125.2 | 541.2 | 161.1 | 407.0 | 352.6 | 205.2 |
| RRT* | 200.4 | 432.0 | 436.5 | 175.3 | 506.4 | 423.7 |
| LazySP | 334.0 | 921.8 | **144.7** | 401.8 | 224.5 | 404.9 |
| **GraphMP** | **124.5** | **192.6** | 211.6 | **59.3** | 116.3 | **95.5** |
| GNN-Explorer | 155.5 | 332.9 | 222.9 | 93.3 | **101.9** | 120.3 |
| **GraphMP with Smoother** | 140.9 | 363.0 | 268.1 | 65.6 | 142.7 | 120.4 |
| GNN-Explorer with Smoother | 175.7 | 529.3 | 293.0 | 107.3 | 136.2 | 150.7 |

computed path. Therefore, we also compare the performance of GraphMP with Smoother and GNN-Explorer with Smoother.

**Ablation study.** Details are reported in the Appendix.

## 4.2 Comparison With Baselines

Experimental results show that our GraphMP achieves 98.6% - 100% success rate across all the environments, being competitive to the state-of-the-art planners (Table 1). Table 2 shows that GraphMP achieves 16.79%, 35.84%, 12.62%, 23.39%, 18.99% and 25.94% shorter paths than GNN-Explorer, on Maze2, UR5, Snake7, KUKA7, KUKA13 and KUKA14, respectively. Furthermore, the GraphMP with the path smoother has the lowest path cost compared to all baselines with a competitive success rate. Table 3 compares the time cost of all the motion planners. Compared to GNN-Explorer, GraphMP finds valid solutions with 19.94%, 42.14%, 5.07%, 36.44% and 20.62% lower time cost on Maze2, UR5, Snake7, KUKA7 and KUKA14, respectively. Overall, the results indicate that GraphMP outperforms all baseline planners by producing high-quality paths with the fastest planning speed and nearly 100% success rate in most of the planning tasks from 2D to 14D.

## 4.3 Study of Individual Modules of GraphMP

**The Performance of Neural Collision Checker.** We first evaluate the neural collision checker by reporting the mean prediction accuracy, confidence score and time cost. Specifically, the mean prediction accuracy and confidence are measured by averaging over all the edges from the test graphs, and the mean time cost represents the latency of predicting all edges per testing graph. Fig. 5 shows that our neural collision checker can perform super fast and accurate prediction on the graph-level collision check with high prediction confidence, in all the environments.

**The Performance of Neural Heuristic Estimator.** We then evaluate the performance of the neural heuristic estimator by comparing the A* with different heuristic functions. To be specific, we equip A* with our neural heuristic estimator and the Euclidean function which serves as the commonly used heuristic function to solve planning tasks, respectively. From Fig. 5, it is shown that the A* with our neural heuristic predictor significantly outperforms the vanilla A*, by producing solutions of lower path cost. The advantage of our neural heuristic predictor also grows quickly with the increasing dimensions of the planning tasks. The results illustrate that the neural heuristic estimator learns and performs better search guidance, especially in the configuration space of higher dimensions.

**The Impact of In-Search Collision Check (ICC) and Lazy Node Removal (LNR).** We further analyze the impact of the ICC and LNR modules on GraphMP, to justify their necessity in improving

Table 4: The comparison of the path cost and time cost (ms) between different versions of GraphMP. To be specific, the GraphMP w/o ICC uses accurate collision check only. GraphMP achieves the significantly lower path cost compared to the one without LNR with a limited time increase, and achieves reduced time cost compared to the one without ICC.

| | Maze2 | | UR5 | | Snake7 | | KUKA7 | | KUKA13 | | KUKA14 | |
|---|---|---|---|---|---|---|---|---|---|---|---|---|
| | Path | Time | Path | Time | Path | Time | Path | Time | Path | Time | Path | Time |
| GraphMP w/o LNR | 2.82 | **106.75** | 9.76 | **183.84** | 5.94 | **208.27** | 8.83 | **54.57** | 15.82 | 104.29 | 14.32 | **88.18** |
| GraphMP w/o ICC | 2.33 | 167.85 | 8.09 | 258.27 | 5.68 | 286.54 | 7.01 | 104.94 | 13.57 | 162.44 | 12.24 | 137.70 |
| **GraphMP** | **2.33** | 124.47 | **8.09** | 192.55 | **5.68** | 211.66 | **7.01** | 59.34 | **13.57** | 116.29 | **12.24** | 95.47 |

the planning performance. Table 4 compares GraphMP with the version without ICC and the one without LNR, with respect to the path cost and time cost, respectively. While all these planners achieve 98.6% - 100% success rate across different environments, GraphMP outperforms the version without LNR by yielding 17.38%, 17.11%, 4.38%, 21.92%, 14.22% and 14.53% shorter paths with a limited increase of the time cost, on Maze2, UR5, Snake7, KUKA7, KUKA13 and KUKA14, respectively. Besides, compared to the one that uses accurate collision check (GraphMP w/o ICC), GraphMP achieves the same path quality with a significantly smaller time cost. Overall, it is seen that ICC and LNR are effective in improving planning speed and path optimality, respectively.

## 5 Limitations

Despite its good empirical performance across different tasks, GraphMP still has some limitations. First, it does not provide probabilistic completeness when the collision check threshold $\theta < 100\%$. That means, if the collision status of some edges is determined by the neural collision checker (NCC), even if the prediction accuracy of NCC is high and a collision-free path exists in the input RGG, GraphMP still cannot guarantee to find the feasible solution asymptotically. Notice that though the probabilistic completeness can be achieved when setting $\theta = 100\%$, the planning time will accordingly increase due to the extra costs incurred by performing accurate collision check on all the explored edges. Second, it does not offer asymptotical optimality. GraphMP performs the graph search on an implicit RGG which is incrementally expanded with more batches of nodes. Once a path is found, GraphMP validates its legality and returns the solution. Because 1) this mechanism naturally leads that the quality of the sampled RGGs has a heavy impact on the path cost – the waypoints along the paths are restricted to be a subset of the existing nodes of RGG, but RGG itself cannot be guaranteed to contain the optimal path; and 2) the search process will be terminated once the path is found, without further seeking better solutions, GraphMP cannot theoretically guarantee to find the optimal path asymptotically. Third, its efficiency is still limited by inefficient RGG construction. More specifically, 1) the uniform sampling of nodes disregards the environmental topology, causing some unnecessary node exploration; and 2) the construction of raw edges is also involved with unnecessary edge generation, thereby limiting the further runtime speedup provided by the proposed approach.

## 6 Acknowledgement

This work is partially supported by National Science Foundation under Grant CCF-2239945, National Science Foundation CPS and SaTC programs.

## 7 Conclusion

This paper proposes GraphMP, a neural motion planner for both low and high-dimensional planning tasks. With the customzied model architecture and training mechanism design, GraphMP can efficiently learn the graph pattern and process graph search, enabling strong planning performance. Experiments on a variety of environments show that GraphMP achieves significant planning speed-up and higher path quality than the state-of-the-art learning-based and classical planners.

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
