# Appendix to "GraphMP: Graph Neural Network-based Motion Planning with Efficient Graph Search"

Xiao Zang[1]    Miao Yin[2*]    Jinqi Xiao[1]    Saman Zonouz[3]    Bo Yuan[1]

[1]Department of Electrical and Computer Engineering, Rutgers University
[2]Department of Computer Science and Engineering, The University of Texas at Arlington
[3]School of Cybersecurity and Privacy, Georgia Institute of Technology
[1]{xz514, jx257, bo.yuan.ece}@rutgers.edu
[2]miao.yin@uta.edu   [3]szonouz6@gatech.edu

## 1   Neural Collision Checker

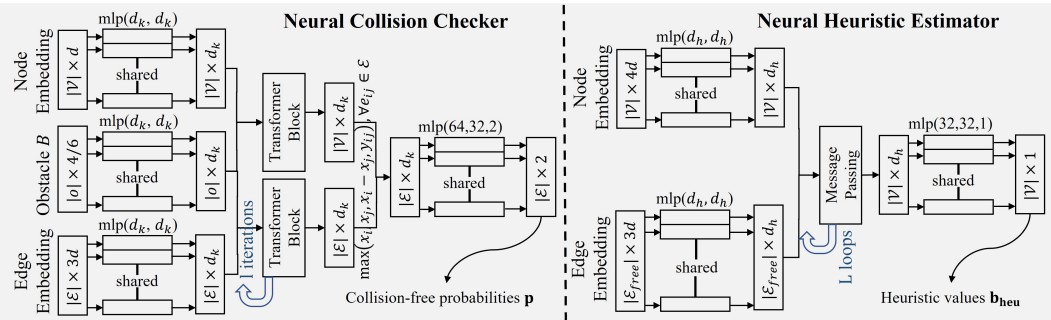

Figure 1: The overall network architectures of the neural collision checker and neural heuristic estimator in GraphMP.

The overall network architecture is shown in Fig. 1. We follow the same way of obstacle encoding from [1]. The initial embedding $x_i^{(0)}$ of each node $v_i$ is set as the corresponding free state and $y_{ij}^{(0)}$ is initialized by concatenating the vectors $(x_i^{(0)}, x_j^{(0)}, x_i^{(0)} - x_j^{(0)})$. For simplicity of notation, we use $x$ and $y$ to represent the embedding of all nodes and edges, respectively. Given MLPs $f_{a_x}^{(t)}$, $f_{a_y}^{(t)}$, $f_{K_x}^{(t)}$, $f_{K_y}^{(t)}$, $f_{Q_x}^{(t)}$, $f_{Q_y}^{(t)}$, $f_{V_x}^{(t)}$, $f_{V_y}^{(t)}$, the obstacle encoding at the $t$-th iteration in the iterative obstacle encoding $f_{oe}$ can be formulated as:

$$a_x^{(t)} = \text{LN}(x^{(t)} + Att(f_{K_x}^{(t)}(B), f_{Q_x}^{(t)}(x^{(t)}), f_{V_x}^{(t)}(B))),$$
$$a_y^{(t)} = \text{LN}(y^{(t)} + Att(f_{K_y}^{(t)}(B), f_{Q_y}^{(t)}(x^{(t)}), f_{V_y}^{(t)}(B))),$$
$$x^{(t+1)} = \text{LN}(a_x^{(t)} + f_{a_x}^{(t)}(a_x^{(t)})),$$
$$y^{(t+1)} = \text{LN}(a_y^{(t)} + f_{a_y}^{(t)}(a_y^{(t)})). \tag{1}$$

Here, LN is the layer normalization [2] and $t$ starts from zero to $l - 1$. Besides, $Att(K, Q, V) = \text{softmax}(QK^T/\sqrt{d_k})V$ where $K \in \mathbb{R}^{n \times d_k}$, $Q \in \mathbb{R}^{m \times d_v}$ and $V \in \mathbb{R}^{n \times d_k}$ are the keys, queries and

---

*This work was done when the author was with Rutgers University.

37th Conference on Neural Information Processing Systems (NeurIPS 2023).

values, respectively. Additionally, $f_{prob}$ is a three-layer MLP with an output dimension of 2. In our implementation, the last MLP $f_{prob}$ is composed of three fully connected layers whose output dimensions are 64, 32 and 2, respectively. We also apply the ReLU activation after its first and second layers, and LogSoftmax after its last layer. The hidden dimension $d_k$ is set as 64.

## 2 Neural Heuristic Estimator

The overall network architecture is shown in Fig. 1. Given the associated robot state $x_i^{(0)}$, we initialize the node embedding of $v_i$ by incorporating the difference and L2 distance to the goal node $v_g$, and initialize the edges using the information of its connected nodes, i.e.,

$$q_i^{(0)} = h_x(x_i^{(0)}, x_g^{(0)}, x_i^{(0)} - x_g^{(0)}, (x_i^{(0)} - x_g^{(0)})^2), \forall v_i \in \mathcal{V},$$
$$r_{ij}^{(0)} = h_y(x_i^{(0)}, x_j^{(0)}, x_j^{(0)} - x_i^{(0)}), \forall e_{ij} \in \mathcal{E}_{free}, \tag{2}$$

where functions $h_x$ and $h_y$ are the two-layer MLPs that embed $v_i$ and $e_{ij}$ into the latent space with $q_i^{(0)} \in \mathbb{R}^{d_h}$ and $r_{ij}^{(0)} \in \mathbb{R}^{d_h}$, respectively. In our implementation, the last MLP $f_{val}$ that outputs the heuristic values is composed of three fully connected layers whose output dimensions are 32, 32 and 1, respectively. We also apply the ReLU activation after its first and second layers. The hidden dimension $d_h$ is set as 32.

**Loss Function of End-to-End Training** The neural heuristic estimator is trained on batches of graph search problem instances $\{(\mathcal{V}^{(i)}, \mathcal{E}_{free}^{(i)}, v_s^{(i)}, v_g^{(i)}, \hat{\mathbf{c}}^{(i)})\}$, where $\hat{\mathbf{c}}^{(i)}$ is a binary vector that marks the nodes contained in the optimal path as one. Similar to [3], for the $i$-th training problem, the training loss is measured by the L1 distance between the closed-list vector $\mathbf{c}^{(i)}$ computed by the differentiable A* and the binary vector $\overline{\mathbf{c}}^{(i)}$ representing the oracle path: $\mathcal{L}_{heu}(\mathbf{c}^{(i)}, \overline{\mathbf{c}}^{(i)}) = ||\mathbf{c}^{(i)} - \overline{\mathbf{c}}^{(i)}||_1 / |\mathcal{V}^{(i)}|$. Such a loss penalizes the nodes that are excessively explored to compute a path and forces the path to be close to the optimal path. Consequently, the loss encourages our differentiable A* to obtain the optimal path with the smallest search effort, forcing neural heuristic estimator to learn and produce better search guidance.

---

**Algorithm 1** Graph-based A* search.

---

**Input:** The weighted graph $\mathcal{G} = (\mathcal{V}, \mathcal{E})$, start node $v_s$, goal node $v_g$
**Output:** The path $\pi$
 1: Initialize the open list $\mathcal{O} \leftarrow \{v_s\}$ and closet list $\mathcal{C} \leftarrow \emptyset$.
 2: Initialize $g(v) \leftarrow 0, \forall v \in \mathcal{V}$.
 3: **while** $v_g \notin \mathcal{C}$ **do**
 4:     Select $v_{sel} \in \mathcal{O}$ with the minimum $g(v) + h(v)$.
 5:     Update $\mathcal{O} \leftarrow \mathcal{O} \setminus v_{sel}, \mathcal{C} \leftarrow \mathcal{C} \cup v_{sel}$.
 6:     **for** $v_{nbr} \in \mathcal{N}(v_{sel}) \cap \overline{\mathcal{C}}$ **do**
 7:         # Here, $w_{v_{sel}, v_{nbr}}$ denotes the weight of edge $e_{v_{sel}, v_{nbr}}$.
 8:         Compute $cost \leftarrow g(v_{sel}) + w_{v_{sel}, v_{nbr}}$.
 9:         **if** $v_{nbr} \notin \mathcal{O}$ or $cost < g(v_{nbr})$ **then**
10:             Update $\mathcal{O} \leftarrow \mathcal{O} \cup v_{nbr}, g(v_{nbr}) \leftarrow cost, p(v_{nbr}) \leftarrow v_{sel}$.
11:         **end if**
12:     **end for**
13: **end while**
14: # Traverse the ancestors of goal $v_g$ until $v_s$ is reached.
15: $\pi \leftarrow Backtrack(p, v_s, v_g)$.
16: **Return** $\pi$.

---

## 3 Admissibility and Consistency.

The A* search is only guaranteed to find the optimal path if the heuristic values are both admissible and consistent. Therefore, it is interesting to study such two properties of our predicted heuristic values produced by NHE. Empirical evaluations show that NHE exhibits admissibility and consistency. Denote $f(v_i)$ be the heuristic value of node $v_i$ and $c(v_i, v_j)$ be the actual cost between $v_i$ and $v_j$. As

shown in the left figures (heatmaps) from Fig. 2(a) and Fig. 2(b), the value $|f(v_i) - f(v_j)| - c(v_i, v_J)$ for each edges $e_{ij} \in \mathcal{E}$ is non-positive for most cases, indicating good consistency for NHE. The right figures in Fig. 2(a) and Fig. 2(b) also show that $f(v_i)$ estimated by NHE is constantly smaller than actual cost $c(v_i, v)g)$, indicating good admissibility of NHE.

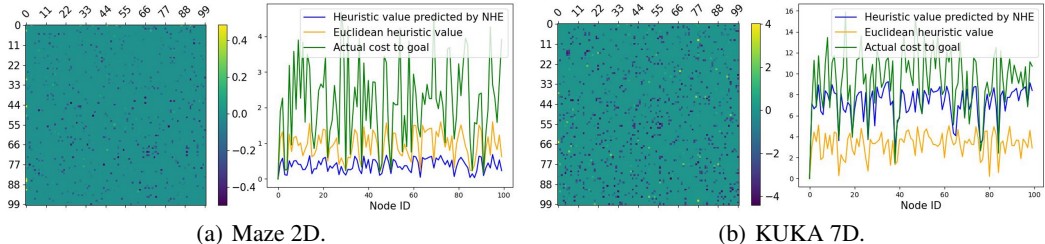

(a) Maze 2D.            (b) KUKA 7D.

Figure 2: Study of consistency and admissibility of the neural heuristic estimator (NHE) on RGG of 100 nodes, on Maze 2D and KUKA 7D, respectively. Denote $f(v_i)$ be the heuristic value of node $v_i$ and $c(v_i, v_j)$ be the actual cost between $v_i$ and $v_j$. The left figures of (a) and (b) show the value of $|f(v_i) - f(v_j)| - c(v_i, v_j)$ for each edge $e_{ij}$ in the RGG, where a negative value (dark color) indicates that the consistency holds for the corresponding pair of nodes. It is seen that NHE exhibits **good consistency**. The right figures in (a) and (b) plot the actual cost to the goal (green line), heuristic values estimated by NHE (blue line) and Euclidean distance (yellow line) of all nodes, respectively. It is seen that NHE exhibits **good admissibility**. To be specific, NHE yields heuristic values that are constantly smaller than the actual cost. In especial, it outperforms the Euclidean distance on KUKA 7D significantly, by approximating the actual cost much better.

## 4 Graph-based A* Search.

A* searches for the minimum-cost path from the start node to the goal by iteratively expanding nodes according to the priority measure: $f(v) = g(v) + h(v)$. Here $g(v)$ is the function that accumulates the actual cost of the path from the start node to $v$, and $h(v)$ is a properly designed heuristic function that estimates the cost from $v$ to the goal node.

Algorithm 1 overviews the procedure of graph-based A* search. The input contains the graph $\mathcal{G} = (\mathcal{V}, \mathcal{E})$, start node $v_s$ and goal node $v_g$. Let $\mathcal{O}$ and $\mathcal{C}$ be the open list (set of nodes to be visited) and closed list (set of nodes that should not be visited again), respectively. Besides, $p(v)$ represents the parent node of $v$. We use $\mathcal{N}(v) = \{v_i | e_{v,v_i} \in \mathcal{E}\}$ to represent the neighboring nodes of $v$. There are two main steps in A*: (1) select the lowest-cost node $v_{sel}$ with the smallest $f(v)$ value from the open list $\mathcal{O}$ (Line 4), and (2) expand the neighborhood of the $v_{sel}$ to update $\mathcal{O}$, accumulated cost $g$ and the parent node (Line 6-13). Specifically, for each valid node $v_{nbr} \in \mathcal{N}(v_{sel})$ that is not in the closet list (Line 6), we check whether $v_{nbr}$ is a first-time seen node or there exists a shorter path to connect it through $v_{sel}$. If so, we append this node $v_{nbr}$ into $\mathcal{O}$, update its cost $g(v_{nbr})$ and set its parent as $v_{sel}$ (Line 10). We repeat the above procedure until the goal $v_g$ is reached, and then retrieve the ordered path by backtracking the ancestor nodes of $v_g$ recursively (Line 15).

## 5 Ablation Study

**Varying Threshold $\theta$ in In-Search Collision Check.** We also investigate the impact of different values of $\theta$, which controls the probability threshold of performing accurate collision check, on our in-search collision check. A smaller value of $\theta$ causes a higher chance to query the predicted collision results but with more uncertainty of correctness; while the greater value brings more latency incurred by accurate collision check but with more safety of being collision-free. From Fig. 3, it is seen that larger $\theta$ brings higher time cost due to the increasing demand of accurate collision checks. On the other hand, the success rate first increases with larger $\theta$, since smaller $\theta$ causes more aggressive approximated collision check, potentially bringing collision-existed solution. Considering when $\theta$ is approaching 80%, the success rate becomes steady across different environments. Therefore, we adopt $\theta$=80% in our experiments to make good balance between planning speed and success rate.

Table 1: The comparison between A* with the heuristic function of neural heuristic estimator (NHE) and vanilla A*, with respect to the mean path cost and search space. The search space is measured by counting the number of visited nodes to compute a path. Note that we do not perform lazy node removal to reduce path cost here. The A* with our neural heuristic predictor significantly outperforms the vanilla A*, by producing solutions of lower path cost with smaller search space.

|  | A* with NHE | | Vanilla A* | |
|  | Path cost | Search sp | Path cost | Search sp |
| --- | --- | --- | --- | --- |
| Maze2 | 2.82 | **19.40** | **2.71** | 23.59 |
| UR5 | **9.76** | **3.65** | 11.38 | 5.97 |
| Snake7 | **5.94** | **3.15** | 6.45 | 7.24 |
| KUKA7 | **8.83** | **7.13** | 10.81 | 15.33 |
| KUKA13 | **15.82** | **8.74** | 18.51 | 23.76 |
| KUKA14 | **14.32** | **7.25** | 17.19 | 24.82 |

Table 2: The performance of neural collision checker on 1000 raw RGGs with 300 nodes and $K$-value of 20 (K-NN).The GNN predicts the collision status of all edges in parallelism.

|  | Accuracy (%) | Confidence (%) | Time cost per graph (ms) |
| --- | --- | --- | --- |
| Maze2 | 98.90±0.66 | 81.62±5.15 | 4.50±0.13 |
| UR5 | 95.31±0.38 | 85.61±5.57 | 5.24±0.09 |
| Snake7 | 96.38±0.41 | 76.98±5.28 | 5.36±0.06 |
| KUKA7 | 94.33±0.83 | 77.26±4.91 | 5.32±0.08 |
| KUKA13 | 93.75±1.57 | 76.60±5.79 | 5.27±0.11 |
| KUKA14 | 91.89±1.21 | 75.04±4.35 | 5.31±0.06 |

**The number of nodes per sampling.** Fig. 4 compares the performance of GraphMP with different numbers of nodes per sampling. We set the max budget of sampled nodes as 1000 and fix the $K$ value of k-NN as 10. It is seen that path cost is reduced slightly with the increasing number of nodes because the denser graphs contain optimal paths with smaller path costs. However, the operation of lazy node removal also fixes the lousy paths caused by the graph sparsity to some degree. Besides, the time cost becomes the lowest when the number of nodes equals 100 but keeps growing with larger numbers. The reason is that sampling nodes from the free space and the computation of K-NN are time-consuming, both too few or too many numbers cause the unnecessary repeat of these operations. Furthermore, the success rates peak when the number is 100 and become stable.

**The $K$ value of K-NN.** Furthermore, we inspect the impact of different $K$ values adopted in K-NN, by fixing the number of nodes per sampling as 100. From Fig. 5, our GraphMP first achieves a smaller path cost with the greater $K$ because the denser graph edges increase the possibility of composing a shorter path. Besides, the time cost decreases when $K = 10$ and then becomes stable. The reason is

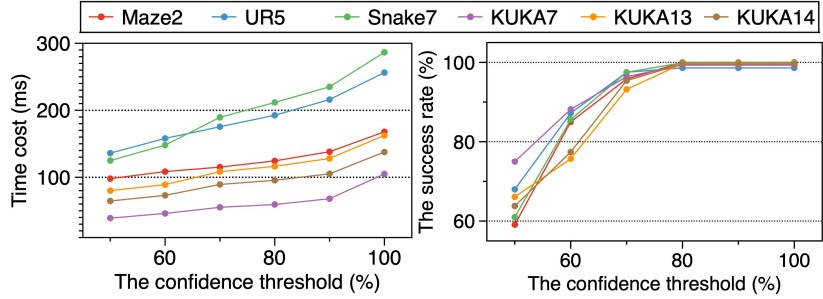

Figure 3: The time cost and success rate with the varying values of confidence threshold $\theta$ in in-search collision check. The success rate is measured by the ratio of testing problems solved by the collision-free paths validated by accurate collision check.

Table 3: The ratio (%) of edges whose prediction confidence are above different thresholds of in-search collision check.

|  | $\theta = 60\%$ | $\theta = 70\%$ | $\theta = 80\%$ | $\theta = 90\%$ |
|---|---|---|---|---|
| Maze2 | 98.83 | 96.99 | 93.98 | 88.73 |
| UR5 | 97.78 | 95.43 | 92.93 | 89.46 |
| Snake7 | 96.14 | 92.62 | 87.32 | 80.16 |
| KUKA7 | 95.51 | 91.60 | 85.75 | 75.46 |
| KUKA13 | 92.26 | 82.39 | 73.32 | 47.78 |
| KUKA14 | 92.54 | 82.98 | 67.10 | 39.94 |

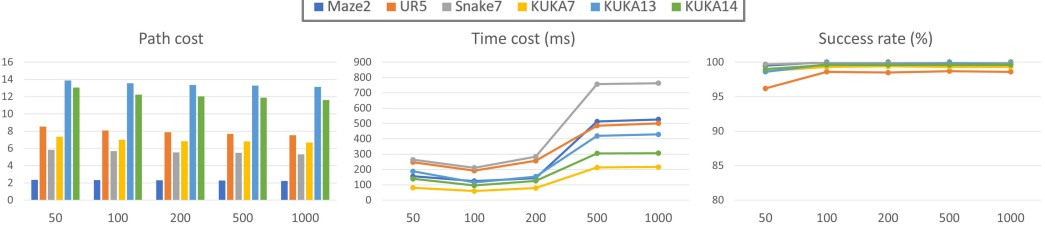

Figure 4: The ablation study on the varying number of nodes per sampling.

that a larger $K$ brings more edges to search a path without additional operations of sampling nodes. The success rate also peaks around $K = 10$.

**The ratio of edges above different values of $\theta$.** Table 3 shows the ratio of edges whose prediction confidence are above different $\theta$. It is seen that most of the raw edges are predicted with the confidence above $\theta = 80\%$ which well preserves the graph structure of the raw RGG, thereby covering the configuration space with good connectivity.

## 6 Probabilistic Completeness of GraphMP

**Lemma 1.** *When the threshold $\theta$ of in-search collision check equals $100\%$, GraphMP is probabilistically complete as more batches of nodes are sampled and added to the RGG.*

*Proof.* The GraphMP performs the accurate collision-check on all visited edges during the graph search procedure, with the threshold $\theta$ of in-search collision check equaling $100\%$. That means the found path must be collision-free as long as a solution is computed. Therefore, proving that GraphMP is probabilistically complete can be done in two steps as follows.

First, the probability that the predicted RGG contains a collision-free path approaches one as more nodes are added, if there exists a valid solution. Given that nodes are sampled uniformly from the free configuration space $\mathcal{X}_{free}$, the probability that a particular node $v \in \mathcal{X}_{free}$ has not been sampled after $i$ batches approaches one, i.e.,

$$\lim_{i \to \infty} (1 - \frac{1}{|\mathcal{X}_{free}|})^{ni}, \tag{3}$$

where $n$ is the number of node per batch. Hence, for any node $v \in \mathcal{X}_{free}$, as the batch increases, $v$ will almost surely be included in the RGG. Assume there exists a valid path comprises of $l$ edges and the RGG is dense infinitely. For each edge $e_{ij}$ of the path, we can find one intermediate node $v_k$ along $e_{ij}$ so that both $e_{ik}$ and $e_{kj}$ are predicted as collision-free correctly, as long as the prediciton accuracy of the neural collision checker is positive. Therefore, the predicted RGG must contain a collision-free path if the solution exists, when enough nodes are sampled.

Second, if there exists a collision-free path among the RGG containing collided edges, the A* equipped with accurate collision checker will find the valid path. Given a RGG $\mathcal{G} = (\mathcal{V}, \mathcal{E})$, A* defines the cost function as $f(v) = g(v) + h(v)$ for each node $v \in \mathcal{V}$, where $g(n)$ is the actual cost

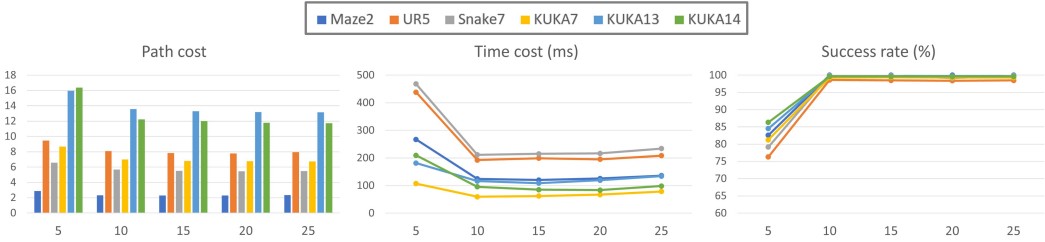

Figure 5: The ablation study on the varying $K$ value of K-NN.

from $v_s$ to $v$ and $h(v)$ is the heuristic estimate from $v$ to $v_g$. The A* initializes the search with $v$ in the open list and compute $f(v)$ in each iteration. A* iteratively selects the node $v_{sel}$ with the lowest $f(v_{sel})$ from the open list for exploration. For every neighboring node $v_{nbr}$ of $v_{sel}$, A* will only append $v_{nbr}$ to the open list if $e_{v_{sel},v_{nbr}}$ is validated by the accurate collision checker. Therefore, any node that can be reached from $v$ through a collision-free path will be appended into the open list. A* terminates when the goal $v_g$ is reached or the open list is empty which contradicts our assumption that there exists a collision-free path. Therefore, the accurate collision checker ensures the A* is able to find the collision-free path if the solution exists.

In nutshell, when the collision checker threshold $\theta = 100\%$, the probability of GraphMP to find a collision-free path approaches one with more batches of sampled nodes. $\square$

## 7  Pseudo Codes

---

**Algorithm 2** In-Search Collision Check (ICC)

---

**Input:** The edge $e_{ij}$, the obstacles $B$, the collision probability $p_{\mathcal{I}(e_{ij})}$, the confidence threshold $\theta$
**Output:** The collision status of $e_{ij}$

1: **if** $p_{\mathcal{I}(e_{ij})} > \theta$ **then**
2:     **Return** True.
3: **else if** AccurateCollisionCheck($e_{ij}, B$) **then**
4:     **Return** True.
5: **end if**
6: **Return** False.

---

---

**Algorithm 3** Lazy Node Removal (LNR)

---

**Input:** The computed path $\pi$, the obstacle $B$
**Output:** The new path $\pi^{'}$

1: **for** $i = 0, 1, 2, ..., \text{len}(\pi) - 1$ **do**
2:     **for** $j = \text{len}(\pi) - 1, \text{len}(\pi) - 2, ..., i + 1$ **do**
3:         **if** AccurateCollisionCheck($\pi_i, \pi_j, B$) **then**
4:             $\pi^{'} \leftarrow [\pi_0, \pi_1, ..., \pi_i, \pi_j, \pi_{j+1}, ..., \pi_{len(\pi)-1}]$.
5:             **Return** LNR($\pi^{'}, B$).
6:         **end if**
7:     **end for**
8: **end for**

---

**Algorithm 4** GraphMP Inference for Online Planning

---

**Input:** Start node $v_s$, goal node $v_g$, the obstacles $B$, the well-trained neural collision checker $f_{col}(\mathcal{V}, \mathcal{E}, B, \mathbf{\Theta_{col}})$ and neural heuristic estimator $f_{heu}(\mathcal{V}, \mathcal{E}, v_g, \mathbf{\Theta_{heu}})$, the confidence threshold $\theta$
**Output:** The collision-free path $\pi$

1: Sample a set of $n$ nodes from $\mathcal{X}_{free}$ as $\mathcal{V}$.
2: Build the graph $\mathcal{G} = (\mathcal{V}, \mathcal{E})$.
3: **repeat**
4:      Predict collision probabilities $\mathbf{p} \leftarrow f_{col}(\mathcal{V}, \mathcal{E}, B, \mathbf{\Theta_{col}})$.
5:      Retrieve collision-free edges $\mathcal{E}_{free} \leftarrow \{e_{ij} | p_{\mathcal{I}(e_{ij})} > 0.5\}$.
6:      Predict heuristic values $\mathbf{b_{heu}} \leftarrow f_{heu}(\mathcal{V}, \mathcal{E}_{free}, v_g, \mathbf{\Theta_{heu}})$.
7:      Search path $\pi \leftarrow$ A*_with_ICC$(\mathcal{V}, \mathcal{E}_{free}, B, v_s, v_g, \mathbf{b_{heu}}, \mathbf{p}, \theta)$.
8:      **if** $\pi == \emptyset$ **then**
9:          Sample another $n$ nodes from $\mathcal{X}_{free}$ and add to $\mathcal{V}$.
10:         Re-compute the edges $\mathcal{E}$.
11:     **end if**
12: **until** $|\mathcal{V}|$ exceeds the max budget of nodes
13: # Perform accurate collision check on edges composing the path.
14: **if** AccurateCollisionCheck$(\pi, B)$ **then**
15:     **Return** $\emptyset$.
16: **else**
17:     # Remove the lazy nodes from the computed path.
18:     Update path $\pi \leftarrow$ LNR$(\pi, B)$.
19: **end if**
20: **Return** $\pi$.

---