# OpenReview forum: "GraphMP: Graph Neural Network-based Motion Planning with Efficient Graph Search"
_NeurIPS.cc/2023/Conference — NeurIPS 2023 poster_

### Official Review · Reviewer_Akwa · 2023-07-02

**Soundness:** 3 good
**Presentation:** 3 good
**Contribution:** 3 good
**Rating:** 7
**Confidence:** 4

**Summary:**

The paper studies learning-based motion planners with GNNs. To improve the performance of GNN-based previous work, this paper proposes the GraphMP, possibly applied to low and high-dimensional planning tasks. The important idea is using (1) using predicted heuristic values, (2) NCC (Neural collision checker), and (3) differentiable A* modules (encoded with matrix operations) for efficient motion planning. The experimental results in Sec.4 showed that the proposed GraphMP performs better with limited computational times.

**Strengths:**

- The proposed concept using multiple components (e.g., learning modules, NCCs, differentiable A*) is clearly explained.  Figure 3 and Algorithm 1 are good for following the concept.
- The solid experimental results in various motion planning domains.

**Weaknesses:**

- I have no ideas except some assumptions (the input RGC is given and how to set the upper bound Tmax for unfamiliar environments).

**Questions:**

I can follow the concept and descriptions in the submitted version, and feel that the contributions are solid. Here are some comments and questions to clarify the paper.

- (Related to the curse of dimensionality?): In the results in Table 1, KUKA7 seems to have a bit worse success rate than those KUMA14. Do you have any reasons for them? I feel that the problems having a larger DoF are harder than those with a lower DoF.
    - Regarding this point, Table 2 and Table 3 are reasonable for me because they have more DOFs. However, only for the success rates in Table 1, the results should be explained in my opinion (maybe, they are some random effects? If so, it is better to mention them, e.g., random seeds, trials, etc.).
- (About the environment $G$): In the setting, RGG $G$ seems to be given and updated lazily. Can we learn G itself together with other learnable modules?
    - In the reported results, LNR seems to be effective in refining paths after A* is finished. Since the current LNR seems independent from the learning part, I guess that the performance depends on the given RGG; therefore, I conjecture that learning representations on graphs is an interesting direction (e.g., learning a mask matrix to prune nodes).
- (About the learning setting): How can we set the max transition time $T_\mathrm{max}$ in the end-to-end learning procedure? What happens if $T_\mathrm{max}$ is not set adequately?
    - I feel that putting the larger $T_\mathrm{max}$ is enough to find feasible solutions (i.e., a path that reaches $v_g$), though it may cause some inefficient phenomena in practice. Do you have any evaluations and discussions?
- (Neural heuristic estimators): The comparison is a bit trivial because vanilla A* adopts the fundamental heuristic functions. Do you have any discussions on Maze2 (where Valina A* showed better path costs)? Do you have any findings of learning heuristic values in your domains?
- Other minor comments:
    - I prefer using explicitly an indicator vector $\mathbb{I}[\mathbf{b}_\mathbf{acc} > \mathbf{b}’_\mathbf{acc}]$ to $\mathbf{b}_\mathbf{acc} > \mathbf{b}’_\mathbf{acc}$ for clarity and following other literature in the ML.
    - $\mathbb{1}$ should be with its dimension (e.g., $\mathbb{1}_{|V|}$) for clarity.

**Limitations:**

I have no explicit concerns about the limitations.

---

> ### Author Rebuttal · Authors · 2023-08-09
>
> We sincerely appreciate the reviewer's very constructive comments and suggestions. The following is our response in order of questions and comments raised.
>
> >Q1: KUKA7 vs KUKA14 in Table 1.
>
> Thank you for pointing it out. The reported a bit worse success rate for KUKA7 than KUKA14 is due to random effects. The current result is based on a single random seed -- GraphMP achieves 99.3\% (round to 99\%) and 99.5\% (round to 100\%) rates on KUKA7 and KUKA14, respectively. After averaging the results using four different random seeds, the success rates on KUKA7 and KUKA14 become 99.4\% and 99.2\%, respectively, being consistent with the trend when DoF increases. We will update the results in the revised version by running with different random seeds. We appreciate your valuable comments.
>
> >Q2: Can we learn RGG?
>
> Thank you for the very valuable suggestion. We highly agree with that learning RGG is an interesting research direction. As we discuss in the limitation and future work **(Sec. 2 of Supplementary Material)**, the current RGG generation method is somewhat inefficient with involving unnecessary node and edge generation, bringing unnecessary exploration and affecting planning performance. In our future work, we plan to fully leverage the workspace topology and the spatial information to bias node sampling/edge construction and/or pruning the unnecessary nodes. One idea is to directly learn and predict the RGG with both nodes and edges, e.g., using generative model, conditioning on the workspace and start/goal states. We are cautiously optimistic and believe that this learning-based RGG construction strategy may help improve the performance of motion planners with respect to planning speed and path quality.
>
> >Q3: Setting of $T_{max}$.
>
> Thanks for the question. In our experiments the value of $T_{max}$ is  simply set as the graph size (the number of nodes), and this setting is large enough to find feasible solutions. To improve the efficiency, in practice we also propose an early stopping mechanism, which checks the sums of all the elements of the open list vector $\mathbf{o}$ within each iteration. Here because the binary vector $\mathbf{o}$ marks all the nodes to be explored as ones, once $\mathbf{o}$ sums up to zero, this means there is no more nodes to be explored and hence the search can be early terminated with avoiding inefficiency. We will include the details of this early stopping mechanism in the revised version. Thank you for pointing it out.
>
> >Q4: Learning heuristic values.
>
> Thank you for the comments. Because it is less challenging to design a good heuristic function in low-dimensional task, A* with admissible heuristic function, e.g., Euclidean distance based, can find very high-quality path (optimal or near-optimal) in 2D maze task. As shown in **Fig. R1(a) of Global Response**, in such scenario the heuristic values generated by manual design is more close to the actual cost than the ones  estimated by learning-based method. This explains why vanilla shows slightly lower path cost than A* equipped with NHE in 2D maze tasks. Notice that this slight cost reduction is not free but with more search efforts.  As shown in **Table 1 in Supplementary Material**, A* with NHE requires smaller search space than vanilla A* with very similar path cost; therefore in overall NHE still shows very competitive overall planning performance as compared to manual heuristic function design in 2D Maze. For planning in high-dimensional cases, as illustrated in **Fig. R1(b) of Global Response**, learning-based solution provides better heuristic value estimations than manual design.
>
> In addition, we have some findings for learning the heuristic values. First,
> training with diverse RGGs improves the generalization of NHE. Based on the observations that 1) the input RGG to NHE is filtered by a collision checker, indicating the uncertainty of the node degrees, and 2) the graph size may increase during the planning phase because extra batches of the sampled nodes would be added if a path can not be found, we construct diverse RGGs when preparing the training data. More specifically, we randomize both the graph sizes and $K$ values of KNN edge construction to ensure that NHE can be trained on different graph structures.
>
> Second, it is important to inform all nodes of the goal information. Considering the heuristic value is to estimate the actual cost to the goal, we initialize the node features of each node by concatenating its own features, the goal features, and the distance to the goal features. Such operation is crucial to learn and predict the heuristic values properly.
>
> Third, The number of the loops of the message passing should be properly selected. The iterations of message passing decides how many hops of neighboring information will be seen from each node. It is important for nodes to receive the information of the whole graph so that high-quality heuristic values can be estimated based on the global information. Too few loops are insufficient to receive the global information while too many loops are likely to cause the over-smoothing issue. Therefore, a proper value of the message passing loop should be determined by the graph size and node degrees. In our work, considering the maximum graph size is 1000 and the $K$ value of KNN is 10, we empirically set the value of the message passing loop as 3.
>
> >Q5: Format of indicator and all-one vectors.
>
> Thank you for pointing it out. We will correct these mathematical formats in the updated version.

---

> > ### Comment · Reviewer_Akwa · 2023-08-17
> > **Thank you for your updates.**
> >
> > I appreciated the contributions of the authors through this discussion and rebuttal phases.
> >
> > The responses above clarified some concerns and my questions in the reviewing process, and I am convinced that the submitted contribution is technically solid and interesting. As you see, I initially rated the paper as 7 (Accept), and so I hope this paper is included in the conference proceeding.

---

> > > ### Author Response · Authors · 2023-08-18
> > > **Thank you for your response**
> > >
> > > Thanks again for the valuable feedback and recognition of our work! We will be sure to update the manuscript according to your suggestions.

---

### Official Review · Reviewer_Dhw9 · 2023-07-05

**Soundness:** 3 good
**Presentation:** 3 good
**Contribution:** 3 good
**Rating:** 7
**Confidence:** 3

**Summary:**

This paper presents an improved graph searching technique, called GraphMP. The algorithm consists of two modules, the neural collision checker and the neural heuristic estimator, which are utilised by a differentable graph-based A* module for path planning.

**Strengths:**

1. The algorithm is presented in detail, supplemented by toy-case studies shown in Fig.1 and Fig.2. They are illustrative.
2. The structure of Secion 3 is good. Each module is explained.
3. The proposed algorithm is compared to multiple existing algorithms, including non-learning-based algorithms.

**Weaknesses:**

1. I think Section 3.1 needs be further polished. Currently, it only states the problem to be solved. Necessary knowledge about how prior works finished the task, and which parts in the existing framework are to be improved are missing. (I can see some of the information in the supplementary file. But I suggest put them in the main text. )

2. Table 1 is not illustrative. The algorithms such as BIT* and RRT* have probabilistic completeness, hence the success rate is high if the computational time is sufficiently long. Currently we cannot appreciate the goodness of the proposed algorithm regarding the completeness. I suggest the authors present the set time limit for the experiments in Table 1 (I didn't find it in Section 4.2). Then, as such, the authors should reduce the time limit to its 75% and show the algorithms' performance. Some of the existing algorithms will not have such good results as shown in Table 1 now.

3. Just a minor comments: It looks like in Section 3.1 the problem is stated in the similar form to what RRT* did, but the algorithm is claimed as a searching-based algorithm. This is a little bit strange, because if the RGG is pre-constructed before running the path searching algorithm, then the presentation can safely start from "give a graph, ..".

4. I didn't understand the "lazy node removal" module. Why is it required? The performance of the algorithm without LNR is not explained well in Section 4.3.

5. The meaning of the terms "w/" and "w/o" are not intuitive. I suggest change them.


**Questions:**

I hope the authors can answer my comments above, and tell me how they will improve the manuscript.

**Limitations:**

I don't think the paper has a potential negative societal impact. I hope the authors can summarise the limitation and briefly talk about the future of their work.

---

> ### Author Rebuttal · Authors · 2023-08-09
>
> We sincerely appreciate the reviewer's very constructive comments and suggestions. The following is our response in order of questions and comments raised.
>
> >Q1. Polishing Section 3.1.}
>
> Thanks for the valuable comment. Given an RGG $G=(V,E)$, source node $v_s$ and goal node $v_g$, the prior work GNN-Explorer initializes an exploration tree $T$ rooted from $v_s$, and visits edges $e_{free} \in E$ sorted by their predicted priorities. Each visit performs the accurate collision check on the edge and tries to append it into $T$, until $v_g$ is reached. Here the search of GNN-Explorer relies on the predicted edge priorities without fully exploring graph structure and formally consider impact of accumulated path cost during path exploration, thereby hampering the path quality. On the other hand, graph-based A* computes the path from the source node $v_s$ via iteratively selecting the best candidate $v_{sel} = argmin_{v \in O} (g(v) + f(v))$ from the open list $O$. Here, $g(v)$ accumulates the actual cost from $v_s$ to $v$, and $f(v)$ estimates the heuristic value of the cost from $v$ to goal $v_g$. Once $v_{sel}$ is selected, each of its reachable neighboring nodes $v_{nbr} \in \mathcal{N}(v_{sel})$ is checked and updated. By using this way, A* can better explore graph structure and perform cost-aware search, but it requires manual design of high-quality heuristic function $h(v)$, which is a challenging task for high-dimensional tasks. Also, visiting the neighboring nodes can be time-consuming because the accurate collision check must be performed on each edge $e_{v_{sel}, v_{nbr}}$. To overcome these limitations of prior works, GraphMP proposes to use GNN to extract and learn the important patterns of RGG, and then identifies the near-optimal path using learnable graph search component. More specifically, a neural collision checker and a neural heuristic estimator are proposed to extract key graph information from input RGG and provide it to the proposed reformulated differentiable A* module for end-to-end training. Therefore, the path planning is now a graph structure-aware and cost-aware process with powerful graph pattern extraction capability and learnable heuristics function, making it can achieve high planning performance and suitable for both low and high-dimensional tasks. We will update Section 3.1 to include this discussion and analysis.
>
>
> >Q2: The time limit of planning.
>
> Thank you for the valuable suggestion. Following the same setting used in Graph-Explorer, in our experiments the time budget is in the format of maximum number of the sampled nodes (as $1000$) for all the planners. By taking your suggestion, we report the success rates of all planners by setting the time limit using wall-clock time (as 400 ms and 300 ms) in the table below. It is seen that under the same wall-clock time budget, GraphMP achieves the highest success rates in almost all the tasks.
>
> **Table R2. The success rates when the time limit is 400 ms and 300 ms, respectively. Results for 300 ms are shown in the parenthesis.**
> |              | Maze2       | UR5         | Snake7      | KUKA7       | KUKA13      | KUKA14      |
> |--------------|-------------|-------------|-------------|-------------|-------------|-------------|
> | GraphMP      | 98.5 (97.1) | 91.1 (88.4) | 99.3 (99.3) | 98.9 (98.6) | 96.1 (95.8) | 97.5 (97.2) |
> | GNN-Explorer | 98.2 (96.6) | 86.3 (81.7) | 99.2 (99.1) | 98.3 (98.2) | 96.4 (95.9) | 97.2 (96.9) |
> | RRT*         | 61.7 (53.3) | 27.8 (14.8) | 47.5 (28.0) | 80.0 (79.6) | 57.2 (49.9) | 65.3 (58.1) |
> | BIT*         | 90.9 (83.4) | 80.9 (76.1) | 98.9 (98.5) | 97.8 (97.1) | 96.1 (94.3) | 95.1 (94.6) |
> | LazySP       | 92.7 (91.7) | 83.3 (78.5) | 99.3 (99.3) | 98.4 (97.6) | 97.6 (97.1) | 97.4 (96.5) |
>
> >Q3: Starting problem statement in Section 3.1.
>
> Thanks for point it out. We will follow your suggestion to revise this part.
>
> >Q4: Lazy node removal.
>
> Thank you for the question.
> The lazy node removal (LNR) aims to reduce path cost by eliminating redundant nodes along the computed path. Specifically, LNR repeatedly attempts to connect each pair of disconnected nodes from both ends of the path via a collision-free edge. All the intermediate waypoints between the connectable nodes will be removed. **Fig. 4(a)** shows an example. The LNR detects that node 0 and 2 can be connected directly, so node 3 is pruned from the path. Our experiments show that LNR can reduce the average path cost by 21.92\% at most on various types of planning tasks. As shown in Table 4, GraphMP without using LNR can achieves faster planning speed with a bit path cost overhead. So whether to use LNR or not is a design choice depending on the specific performance requirement in the application scenarios (path cost-sensitive or time cost-sensitive).
>
> >Q5: "w/" and "w/o".
>
> Thanks for pointing it out. We will use the full forms in the revised version.
>
>
> >Q6: Limitation and future work.
>
> Thank you for the comments. We discuss the limitation and future work in **Sec. 2 of Supplementary Material**. GraphMP may suffer from the low quality of the raw RGGs due to the uniformly sampled nodes and KNN edges. Such RGG construction strategy neglects the workspace topology and the specific task information, which may waste resources in the exploration of space that is unable to yield optimal paths. Therefore, in our future work, we plan to investigate more efficient graph construction methods, e.g., graph generative model to generate node and edges simultaneously, to construct high-quality RGGs with fewer and only necessary nodes and edges for finding the optimal paths.

---

> > ### Comment · Reviewer_Dhw9 · 2023-08-16
> >
> > Thank you for the rebuttal. I can see most of my questions have been solved.
> >
> > 1. The remaining problem is still about the significance of algorithm. It should be noticed that without the add-on modules (e.g., LNR), the path distance optimality of the proposed algorithm (Table 4, line 1) is not good compared to that of other algorithms (presented in Table 2). This means that the claimed contribution "achieves significant improvement on path quality and ..." in abstract is essentially given by the add-ons, not the graph-based structure itself. In other words, if I add a smoother to BIT*, the resultant path length of BIT* will also improve.
> >
> > 2. The other issue I want to mention after reading other reviewers' comments is that, there should be a formal discussion about the completeness (is the proposed algorithm complete?) and optimality (here it means the optimality without add-ons, because the authors should know that in a simply-connected environment ALL path planner algorithms, after equipped with a shortcut optimiser, are optimal).
> >
> > P.S. Don't make me wrong, I don't mean the algorithm must be both complete and optimal. I don't want to distress the authors.

---

> > > ### Author Response · Authors · 2023-08-18
> > > **Rebuttal by Authors**
> > >
> > > >Q7: Planning performance with and without LNR.
> > >
> > > Thank you for the valuable comments. We believe we cannot directly conclude that GraphMP without using LNR has inferior path quality than other algorithms such as BIT*, via comparing results in Table 2 and Table 4. This is because the results in Table 2 and Table 4 are based on the setting of the same ''maximum number of the sampled nodes (1000)" for all planners (we follow this setting used in GNN-Explorer paper (NeurIPS'21)), instead of ''the same planning wall-clock runtime budget". Consider 1) GraphMP has higher planning speed than others (shown in Table 3) and 2) planning time can impact the path quality, e.g., path cost of BIT* can be reduced with more planning time; therefore, when evaluating GraphMP and BIT* under the same runtime budget instead of same sampled nodes budget, GraphMP, with and without LNR, will show better path quality performance in this setting. As shown in the following Table R3 and Table R4, with 200ms planning time budget,  GraphMP without using LNR provides lower path cost than LNR-free BIT* in most tasks, meanwhile achieving much higher success rates.
> > >
> > > We also evaluate the path costs when both GraphMP and BIT* are equipped with LNR modules. As shown in the following Table R5, using LNR improves the path quality of both planners, and GraphMP shows better path quality performance than BIT* in most tasks when the same 200ms planning time limit is set.
> > >
> > > **Table R3. The mean success rates (\%) when the time limit is 200ms.**
> > > |                     | Maze2 |  UR5 | Snake7 | KUKA7 | KUKA13 | KUKA14 |
> > > |:-------------------:|:-----:|:----:|:------:|:-----:|:------:|:------:|
> > > | GraphMP without LNR |  95.2 | 84.7 |  99.1  |  98.5 |  95.4  |  97.1  |
> > > |   BIT* without LNR  |  73.9 | 71.7 |  96.5  |  96.4 |  90.3  |  91.9  |
> > >
> > >
> > > **Table R4. The mean path cost when the time limit is 200ms.**
> > > |                     | Maze2 |  UR5 | Snake7 | KUKA7 | KUKA13 | KUKA14 |
> > > |:-------------------:|:-----:|:----:|:------:|:-----:|:------:|:------:|
> > > | GraphMP without LNR |  2.37 | 6.85 |  4.97  |  6.71 |  10.26 |  9.93  |
> > > |   BIT* without LNR  |  2.13 | 7.69 |  5.41  |  6.94 |  10.53 |  10.64 |
> > >
> > >
> > > **Table R5. The mean path cost when applying LNR to both GraphMP and BIT\*. The time limit is 200ms.**
> > > |                  | Maze2 |  UR5 | Snake7 | KUKA7 | KUKA13 | KUKA14 |
> > > |:----------------:|:-----:|:----:|:------:|:-----:|:------:|:------:|
> > > | GraphMP with LNR |  2.25 |  6.7 |  4.81  |  6.35 |  10.04 |  9.71  |
> > > |   BIT* with LNR  |  2.12 | 7.35 |  5.26  |  6.64 |  10.3  |  10.35 |

---

> > > > ### Author Response · Authors · 2023-08-18
> > > > **Rebuttal by Authors**
> > > >
> > > > >Q8: More discussion about completeness and optimality.
> > > >
> > > > Thank you for the valuable suggestion. We analyze the completeness and optimality of GraphMP as follows.
> > > >
> > > > **Completeness.** When $\theta$ is set as 100\%, i.e., turn off in-search collision check, meaning all the explored edges will be accurately checked. In such scenario, if there exists a collision-free path in the RGG, GraphMP can always find it given sufficient batch sampling, exhibiting probabilistic completeness. When $\theta$ is set as less than 100\%, meaning the collision status of some edges is determined by the neural collision checker (NCC). In such scenario, probabilistic completeness cannot be theoretically guaranteed. However, because NCC has high prediction accuracy and the legality of the final identified path will be examined using accurate collision check, the property of GraphMP in this setting still closely approaches probabilistic completeness. As shown in Fig. 3 of the Supplementary Material, the success rate with $\theta$=80\% increases and approaches 100\% as the number of samples increases, being consistent with the phenomenon of probabilistic completeness. Overall, the user can select different options of GraphMP according to specific practical needs, e.g., a completeness-ensured solution or pursuing faster planning with still achieving empirically high success rates.
> > > >
> > > >
> > > > **Optimality.** Similar to other state-of-the-art learning-based planners like GNN-Explorer and NEXT [R1], because GraphMP terminates the search process once the path is found, it is not theoretically guaranteed to be asymptotic optimality. To be specific, GraphMP performs the graph search on an implicit RGG which is incrementally expanded with more batches of nodes. Once a path is found, GraphMP validates its legality and returns the solution without further seeking better solutions. Evidently, this mechanism naturally leads that the quality of the sampled RGGs has a heavy impact on the path cost of GraphMP -- the waypoints along the paths are restricted to be a subset of the existing nodes. Therefore, in order to further improve path quality without using smoother, one empirical solution is to learn the construction of a high-quality RGG, where the RGG contains the nodes that can constitute the optimal path. Meanwhile, if aiming to make GraphMP asymptotically optimal theoretically, we believe that we can incorporate the incremental search technique [R2] to the GraphMP, e.g., adding new samples to improve the found path in a continuous way, thereby generating asymptotically optimal paths. In our future work we will explore these two solutions.
> > > >
> > > > [R1] Chen, Binghong, et al. "Learning to plan in high dimensions via neural exploration-exploitation trees." arXiv preprint arXiv:1903.00070 (2019).
> > > >
> > > > [R2] Koenig, Sven, Maxim Likhachev, and David Furcy. "Lifelong planning A*." Artificial Intelligence 155.1-2 (2004): 93-146.

---

> > > > > ### Comment · Reviewer_Dhw9 · 2023-08-18
> > > > >
> > > > > Thank you for the feedback. I generally agree with the authors' reply. Please see my small comments for the Q8.
> > > > >
> > > > > 1. Completeness: The algorithm is claimed to be probabilistic complete, hence a more rigorous proof is recommended. Writing the current paragraph directly into the paper is not elegant.
> > > > >
> > > > > 2. Optimality: Since the algorithm is not probabilistic optimal, the authors might have mixed the discussion of the current work and future work. I understand that this is for rebuttal. When revising the manuscript, please write them separately, in a paragraph which solely states the non-existence of probabilistic optimality, and anothe paragraph which solely states the future work.

---

> > > > > > ### Author Response · Authors · 2023-08-18
> > > > > > **Thank you for your comments and suggestions**
> > > > > >
> > > > > > Thank you again for your very constructive comments! We will update the manuscript following your suggestion.

---

### Official Review · Reviewer_KcNw · 2023-07-06

**Soundness:** 2 fair
**Presentation:** 2 fair
**Contribution:** 2 fair
**Rating:** 7
**Confidence:** 4

**Summary:**

The paper extends the previous GNN-Explorer work and replaces its greedy search strategy with A* search using a neural heuristic. Furthermore, it utilizes the neural collision checker and lazy node removal to improve the success rate and path cost. The result is evaluated based on a benchmark from 2D maze to 14D dual arms and shows that it outperforms the baselines.

**Strengths:**

1. The approach is clear and easy to understand.
2. The extensive results on 6 environments seem thorough and show the soundness of the approach. Especially for UR5, the planning time is fast.
3. The ablation studies show the effectiveness of each key component.

**Weaknesses:**

1. The novelty is incremental. All the key components can be found in previous works, including graph-based search (from GNN-Explorer), neural heuristic and differential training (from neural A*), neural collision checker (from Fastron and ClearanceNet), and lazy node removal (from Motion Planning Networks).
However, combining them and showing validness is also important, therefore, I recommend borderline acceptance.

**Questions:**

1. I wonder how sensitive the algorithm is to the hyperparameter θ. As claimed in the paper, this parameter balances the trade-off between the planning efficiency and the path safety. So could you please give an ablation study on how robust the algorithm is to the choice of θ?

2. Do you mind providing the TP, TF, NP, NF scores for Figure 5 (Left)? I would love to know more about whether this neural collision checker is overestimating or underestimating the collision risk.

3. During training, does the NHE take the graph input that is predicted and filtered by the NCC, or does it completely take a collision-free RGG? If it is the later one, then there might be some distribution shift happening during inference.

**Limitations:**

Please discuss the limitations, since I do not see the texts mentioning the limitations. An obvious limitation is that the algorithm is not complete. For example, it is possible that the neural network predicts an edge to be collision-free with very high confidence (say 90%), and the algorithm could find a path that is actually in collision because of it.

I do not see any noticeable negative societal impact.

---

> ### Author Rebuttal · Authors · 2023-08-06
>
> We sincerely appreciate the reviewer's very constructive comments and suggestions. The following is our response in order of questions and comments raised.
>
> >Q1: Technical novelty.
>
> Thanks for the valuable comments. As we state in the Related Work Section, neural network-enabled motion planning is an active research field in recent years, and different components of planners have been studied from learning-based perspective in the literature. Compared with the existing efforts, GraphMP has several technical contributions.
>
> First, as analyzed in Introduction section, GNN-Explorer essentially uses GNN to identify edge priority to grow the exploration tree. Due to its tree exploration-based planning strategy, the mechanism of GNN-Explorer mainly focuses on finding edge priority (a type of "hard" information). Instead, GraphMP does not use tree-based exploration and the GNN in GraphMP serves to learn the graph information to estimate heuristic values. Therefore, GraphMP fully explores graph structure and properly considers the impact of critical accumulated cost (a type of "soft" information) in the planning phase, providing better planning performance (success rate, path cost and planning speed) than GNN-Explorer.
>
> Second, as mentioned in Section 3.5,  Neural A* is a work focusing 2D planning task. More specifically, it estimates the cost from the source to all nodes instead of learning heuristic values of each node. Also, it uses convolutional layers to process the input image information, limiting to the grid-based 2D planning task. On the other hand, the GNN-based model structure and the differentiable training of GraphMP is designed for graph input and can be applied to planning tasks of any dimensions.
>
> Third, both Fastron and ClearanceNet are designed for predicting whether a single node (robot state/configuration) is in the obstacle space or not. To check the collision status of the edge (robot movement), these two methods need to discrete the edge to a set of points and then examine each point. Instead, GraphMP uses neural collision checker (NCC) to learn the collision status of edges, therefore it can directly check the clearance of all edges in the graph without discretizing edges into points.
>
> Also, we agree that removing redundant nodes/states to improve path quality is a common post-processing practice in the motion planning literature, and the shortcutting methods in our work and MPNet can be viewed as the simplified version of the path smoothing with shortcutting heuristic. This operation makes trade-off between path quality and planning time and can be optionally used if required. We will add the related path smoothing literature and discussion in the updated version.
>
>
> >Q2: Sensitiveness of $\theta$.
>
> Thanks for pointing this out. Ablation study on the impacts of $\theta$ on planning time and success rate are reported in **Fig. 2 of Supplementary Material**. It show that larger $\theta$ brings higher time cost due to the increasing demand of accurate collision checks. On the other hand, the success rate first increases with larger $\theta$, since smaller $\theta$ causes more aggressive approximated collision check, potentially bringing collision-existed solution. Considering when $\theta$ is approaching 80%, the success rate becomes steady across different environments. therefore we adopt $\theta$=80% in our experiments to make good balance between planning speed and success rate. These figures are also reported in **Fig. R3 of Global Response**.
>
> >Q3: TP/FP/TN/FN scores of NCC.
>
> Thanks for the suggestions. The following table R1 reports TPR/FPR/TNR/FNR of NCC for Fig. 5. It is seen that NCC has sufficiently large TPR and TNR and small FPR and FNR, demonstrating good prediction capability. Based on this well-performed NCC, the success rate of GraphMP is 99%-100% across different tasks (reported in **Table 1 of main paper**).
>
> **Table R1. TPR, TNR, FPR, FNR of NCC.**
> |     | Maze2 | UR5   | Snake7 | KUKA7 | KUKA13 | KUKA14 |
> |-----|-------|-------|--------|-------|--------|--------|
> | TPR | 98.84 | 93.15 | 96.41  | 94.29 | 93.73  | 91.86  |
> | TNR | 98.93 | 93.17 | 96.35  | 94.37 | 93.77  | 91.92  |
> | FPR | 1.07  | 6.83  | 3.65   | 5.63  | 6.23   | 8.14   |
> | FNR | 1.16  | 6.85  | 3.59   | 5.71  | 6.27   | 8.08   |
>
> >Q4: Input graph for NHE training.
>
> Thank you for this valuable question. In the training process the input of NHE is the completely collision-free RGGs instead of predicted RGGs from NCC. The reason why we make this design choice is that the predicted RGGs generated by NCC can potentially contain collided edges, and hence using this noisy training data may affect the performance of NHE. Instead, training NHE using completely collision-free RGGs can better help NHE to learn a more suitable heuristics function. As shown in the experimental results, GraphMP can achieve strong planning performance with this input difference for NHE between inference and training phases.
>
> >Q5: Potential Limitation, e.g., final found path may not be collision-free?
>
> Thank you for the comments. In **Sec. 2 of Supplementary Material**, we discuss the limitation of our approach and potential research direction. More specifically, the construction of raw input RGG may be inefficient because of the uniform node sampling and KNN-based edge connection. Without fully leveraging the specific prior spatial information, such straightforward RGG building method is involved with many non-necessary nodes sampling and edge construction, affecting construction efficiency.
>
> For the legality of final found path, GraphMP guarantees that the final path, if found, is collision free. As described in **Algorithm 4 of Supplementary Material**, accurate collision check will be performed on the final computed path. Based on this mechanism and the good prediction performance of NCC, the success rate of GraphMP for finding collision-free paths achieves 99%-100% across different tasks.

---

> > ### Comment · Reviewer_KcNw · 2023-08-19
> > **Thanks for partially addressing the issues**
> >
> > My main concern is Q5. See the corresponding paragraph as below.
> >
> > Thanks for explaining the technical difference in Q1. Now I understand better about how you integrate and modify the previous works as the whole framework.
> >
> > The result from Q2 is reasonable and expected, and thanks for conducting the experiment. One thing you can do in the revision is to add the ratio of the number of edges that is above theta to the whole number of edges, with regard to theta. No need to do it now in the rebuttal, since this is not my main concern.
> >
> > Thanks for providing Q3 result, which shows the NCC is a balanced checker. This is good and addresses my concern.
> >
> > Your answer to Q4 is a little tricky, since I’m asking whether making it as a in-distribution graph will make the current result better (definitely the result from the current setting is already good). This question mainly serves as a suggestion for improvement, it would not affect my criteria for acceptance.
> >
> > Q5 is still a severe problem. I still strongly suggest to incorporate the discussion for incompleteness in the limitation section. The reason that NeurIPS provides this section is to describe the drawback your own approach more thoroughly and objectively. There is just no way to achieve the completeness if the NCC is used, and all you need to do is just to claim it in the limitation section. I would not suggest to reject if you claim such a drawback. However, it would be a severe problem if you realize such a problem (as the discussion to Dhw9) and choose to ignore it in the limitation section.

---

> > > ### Author Response · Authors · 2023-08-20
> > > **Thank you for your feedback and suggestions**
> > >
> > > Thank you very much for reading our responses and providing very constructive feedback. We are happy that we have addressed some of your concerns. Next we describe our response to the remaining concerns.
> > >
> > > >For Q2
> > >
> > > Thank you for the valuable suggestion! We will follow your comments and add the figure showing the ratio of edges with respect to different intervals of prediction confidence in the revised manuscript.
> > >
> > > >For Q4
> > >
> > > Thank you very much for the valuable suggestion, and pointing out the option of training NHE using the output of NCC. As we described in the previous response, the reason why we use collision-free RGG is to avoid the impact of training data containing collided edges. On the other hand, as you suggest, using predicted RGG in both training and inference can potentially further improve the performance because of the benefit of in-distribution. This is really a very good suggestion to remind us to explore different data preparation strategy, and we will follow your suggestion. Due to the limited time in the author-reviewer discussion phase, we are not able to provide the update of training all models using RGG generated by NCC at the current moment. We will keep working on this and add the experimental results using the predicted RGG for training in the updated manuscript.
> > >
> > >
> > > >For Q5:
> > >
> > > Thank you for the very constructive comments! We completely agree that the limitations of a work should be described more thoroughly and objectively. In our previous response to you, we explained that GraphMP will not generate the final path containing collided edges because of performing the legality check before generating the final output, in other words, "GraphMP will not produce the collision-included paths". Now we understand that you are referring to the completeness problem, in other words, "GraphMP cannot theoretically guarantee to find collision-free paths asymptotically." We really appreciate you and Reviewer Dhw9 for pointing out this limitation, as well as the optimality problem, which we did not realize in the original submission. Following the valuable suggestion from you and Reviewer DhW9, we will revise and expand the existing Limitation Section of the paper to incorporate the discussion for incompleteness and non-existence of optimality. A draft version of the updated Limitation Section is prepared as follows.
> > >
> > > **Limitation of This Work**
> > >
> > > Despite its good empirical performance across different tasks, GraphMP still has some limitations as follows.
> > >
> > > First, it does not provide probabilistic completeness when $\theta$<100%. That means, if the collision status of some edges is determined by the neural collision checker (NCC), even if the prediction accuracy of NCC is high and a collision-free path exists in the input RGG, GraphMP still cannot guarantee to find the feasible solution asymptotically. Notice that though the probabilistic completeness can be achieved when setting $\theta$=100% (proof will be included in another section of the supplementary material), the planning time will accordingly increase due to the extra costs incurred by performing accurate collision check on all the explored edges.
> > >
> > > Second, it does not offer asymptotical optimality. GraphMP performs the graph search on an implicit RGG which is incrementally expanded with more batches of nodes. Once a path is found, GraphMP validates its legality and returns the solution. Because 1) this mechanism naturally leads that the quality of the sampled RGGs has a heavy impact on the path cost -- the waypoints along the paths are restricted to be a subset of the existing nodes of RGG, but RGG itself cannot be guaranteed to contain the optimal path; and 2) the search process will be terminated once the path is found, without further seeking better solutions, GraphMP cannot theoretically guarantee to find the optimal path asymptotically.
> > >
> > > Third, its efficiency is still limited by inefficient RGG construction. More specifically, 1) the uniform sampling of nodes disregards the environmental topology, causing some unnecessary node exploration; and 2) the construction of raw edges is also involved with unnecessary edge generation, thereby limiting the further runtime speedup provided by the proposed approach.
> > >
> > > Again, we appreciate Reviewer KcNw very much for the very constructive suggestions and comments.

---

> > > > ### Comment · Reviewer_KcNw · 2023-08-20
> > > >
> > > > Thanks for addressing the questions. I've updated the score to 7.

---

> > > > > ### Author Response · Authors · 2023-08-20
> > > > > **Thank you for your suggestions and positive feedback**
> > > > >
> > > > > Thanks a lot for your valuable comments to improve the quality of this manuscript! We will be sure to update the paper according to your suggestions.

---

### Official Review · Reviewer_3bor · 2023-07-07

**Soundness:** 3 good
**Presentation:** 3 good
**Contribution:** 3 good
**Rating:** 7
**Confidence:** 4

**Summary:**

This paper presents GraphMP, a neural motion planner that uses GNNs and graph search techniques to do motion planning in various scenarios. GraphMP has two components: a neural collision checker that estimates the collision status of edges in a randomly sampled graph, and a neural heuristic estimator that assigns heuristic values to nodes for directing the graph search. This paper also develops a graph-based A* algorithm that allows end-to-end learning of the neural heuristic estimator. Thi paper evaluates GraphMP on six planning tasks with dimensions from 2D to 14D and demonstrates that it performs better than several classical and learning-based planners in terms of path quality, planning speed and success rate.

**Strengths:**

- This paper provides extensive experiments and comparisons with baselines on various environments and metrics. It shows strong experimental results in many planning settings.
- The paper leverages the advantages of both GNNs and graph search algorithms to improve performance over existing methods like GNN-Explorer.
- Effective techniques like in-search differentiable graph-based A*, collision check and lazy nod removal are added to improve the performance of GraphMP.


**Weaknesses:**

- This paper does not study or show how different factors and design choices affect the performance of the proposed GraphMP algorithm, including
  - number of neighbors (K) in K-NN (was set to 10 and 20 in different experiments)
  - the graph size
  - the in-search collision check threshold
- This paper does not mention the challenges or drawbacks of GraphMP, such as how it deals with dynamic or uncertain settings, or how it adapts to more complicated tasks.
- It will be helpful to provide more information about how the neural heuristic estimator and the neural collision checker are trained.


**Questions:**

- For A* to work provably, how to ensure that the neural heuristic estimator produces admissible or consistent heuristic values?
- How to handle non-reachable goal states?
- Can a discussion about different design choices in the algorithm be included?


**Limitations:**

Please include a limitation section

---

> ### Author Rebuttal · Authors · 2023-08-06
>
> We sincerely appreciate the reviewer's very constructive comments and suggestions. The following is our response in order of questions and comments raised.
>
> >Q1: Impact of K, graph size and $\theta$.
>
> Thank you for pointing it out. **Fig. 2-4 in Sec. 6 (Ablation Study) of Supplementary Material** report the impacts of these design choices on performance. Detailed analysis is also discussed in that section. More specifically, for **varying threshold $\theta$** of in-search collision check, evaluation results show that time cost keeps increasing with a greater $\theta$, because more accurate collision check is performed during the graph search; and success rate increases first and then becomes steady when approaching $80\%$. Therefore, we set $\theta$ as $80\%$ to achieve good balance between planning speed and success rate. For **varying $K$ value of KNN**, we report the mean path cost, time cost, and success rate with a fixed batch size of node sampling as $100$. It is seen that the path cost and time cost first decreases and then become stable as $K$ increases. This is because denser graph edges are more likely to compose a shorter path and more connections can improve the exploration efficiency. Meanwhile, success rate first increases and then become stables as $K$ approaches 10. So in our experiments $K$ is set as 10. Note that $K=20$ in Fig. 5 of main paper is only used to show the good performance of NCC, we have updated this figure in **Fig. R2 of Global Response**. For **varying graph size**, we evaluate the performance of GraphMP with respect to different number of nodes per sampling, which corresponds to the incremental graph size. Here the max budget of sampled nodes (maximum graph size) is set as $1000$. It is seen that the time cost becomes the lowest when the number of nodes equals $100$ and then grows with more nodes. This is because too few nodes incur the extra sampling batch, while too many nodes are more than necessary to compute a valid solution. The success rate and path cost achieve the best result when the sampling size is $100$. These figures are also reported in **Global Response (Fig. R3-R5)**.
>
>
> >Q2: Potential limitation, e.g., cannot work in dynamic settings?
>
> Thanks for the valuable comments. The limitation of our approach is analyzed in **Sec. 2 of Supplementary Material**. GraphMP may suffer from inefficient construction (uniform node sampling and KNN-based edge connection) of raw input RGG. Without leveraging workspace topology and spatial relationship between the start/goal states, the raw RGG requires more than necessary nodes and edges to cover the configuration space. Our future work will explore how to directly learn the biased node sampling and construct necessary edges.
>
> We believe it is feasible to extend GraphMP for dynamic changing environment. Considering the obstacle's movements are typically predictable, instead of predicting a single-value collision probability for each edge, NCC can learn to predict a vector of probabilities where each entry represents the probability at the corresponding time window. The A* search then looks up the edge probabilities at each timestamp, and plans over an updated collision-free RGG without heavy collision checks, making GraphMP adaptive for dynamic planning.
>
>
> >Q3: More information on training NHE and NCC.
>
> Thanks for the suggestion. The training data consists of 2000 different workspaces for both NCC and NHE. For each workspace, we randomly construct 20 RGGs by sampling a random number of nodes ([100, 200, 300, 400]) and a random $K$ value of KNN ([5, 10, 15, 20]). Adam optimizer is used for both training of NCC and NHE, and the learning rate, training epoch and batch size are set as $1e^{-3}$, $400$ and $8$, respectively.
>
> **For NCC training**, given the input data as the raw RGG and the ground-truth output as the collision status of all edges, NCC predicts the probabilities of being collision-free for all edges in parallelism, and the binary cross-entropy loss is adopted. **For NHE training**, as shown in **Algorithm 1**, given a collision-free RGG $G = (V, E_{free})$ and start/goal nodes ($v_s$, $v_g$), the ground-truth output is a length-$|V|$ binary vector ($\hat{\mathbf{c}}$), denoting the optimal path computed by Dijkstra. NHE predicts a length-$|V|$ vector $\mathbf{b_{heu}}$ where each entry denotes the heuristic value of the corresponding node. Differentiable A* performs A* search to calcualte $\mathbf{c}$, a length-$|V|$ binary vector that is repeatedly updated by marking the explored nodes as ones. The difference (L1 loss) between $\mathbf{c}$ and $\hat{\mathbf{c}}$ is measured as training loss.
>
> >Q4: Admissibility/consistency of NHE.
>
> Thank you for valuable comments. Empirical evaluations show that NHE exhibits admissibility and consistency. Denote $f(v_i)$ be the heuristic value of node $v_i$ and $c(v_i, v_j)$ be the actual cost between $v_i$ and $v_j$. As shown in the left figures (heatmaps) from **Fig. R1(a)(b) of Global Response**, $|f(v_i) - f(v_j)| - c(v_i, v_j)$ for each edge $e_{ij} \in E$ is non-positive for most cases, indicating good consistency for NHE. The right figures in Fig. R1(a)(b) also show that $f(v_i)$ estimated by NHE is constantly smaller than actual cost $c(v_i, v_g)$ ($v_g$ is goal node), indicating good admissibility of NHE.
>
>
> >Q5: How to handle non-reachable goal states?
>
> Thanks for pointing this out. GraphMP uses batched sampling strategy for the case when goal is not reached. As shown in  **Algorithm 4 of Supplementary Material**, GraphMP first constructs a raw RGG using a single batch of nodes and searches for the collision-free path. If the path is not found, i.e., the goal state is not reached, another batch of nodes are sampled and appended to the graph. The above procedure is repeated until the collision-free path is found or the maximum budget of samples is reached. Accurate collision check is performed  on the final computed path to verify the legality.

---

> > ### Comment · Reviewer_3bor · 2023-08-16
> > **thanks for the rebuttal**
> >
> > I would like to thank the authors for the rebuttal and the additional experiments! Nice work! I have update my score accordingly.

---

> > > ### Author Response · Authors · 2023-08-18
> > > **Thank you for your response**
> > >
> > > Thanks a lot for the constructive comments and positive feedback!

---

### Author Rebuttal · Authors · 2023-08-09

**We would like to thank all reviewers for the valuable comments and suggestions.**  In the attached PDF file, we include five figures as described as follows: the study on the admissibility/consistency of the neural heuristic estimator **(Fig. R1)**, the updated neural collision checker results on RGG when $K = 10$ **(Fig. R2)**. Besides, for the reviewer's convenience, we also put the ablation study on varying $\theta$ **(Fig. R3)**, varying sampling size **(Fig. R4)** and varying $K$ **(Fig. R5)**.

---

### Decision · Program_Chairs · 2023-09-21

**Decision:**

Accept (poster)

**Comment:**

The paper presents GraphMP, an approach for motion planning that uses a graph neural network to accelerate planning. It uses networks to predict both collision and heuristics to accelerate planning and proposes a training mechanism. Results over a variety of planning problems show impressive performance.

The reviewers all agreed the work should be accepted, and I concur. The paper is generally clear and thorough and has been improved through the rebuttal process and clarifications the authors provide.